# Measuring currents, ice drift, and waves from space: the Sea Surface KInematics Multiscale monitoring (SKIM) concept

Fabrice Ardhuin[1], Yevgueny Aksenov[2], Alvise Benetazzo[3], Laurent Bertino[4], Peter Brandt[5], Eric Caubet[6], Bertrand Chapron[1], Fabrice Collard[7], Sophie Cravatte[8], Jean-Marc Delouis[1], Frederic Dias[9], Gerald Dibarboure[10], Lucile Gaultier[7], Johnny Johannessen[4], Anton Korosov[4], Georgy Manucharyan[11], Dimitris Menemenlis[12], Melisa Menendez[13], Goulven Monnier[14], Alexis Mouche[1], Frederic Nouguier[1], George Nurser[2], Pierre Rampal[4], Ad Reniers[15], Ernesto Rodriguez[12], Justin Stopa[1], Celine Tison[10], Clement Ubelmann[15], Erik van Sebille[16], and Jiping Xie[4]

[1]Laboratoire d'Océanographie Physique et Spatiale (LOPS), Univ. Brest, CNRS, Ifremer, IRD, Brest, France
[2]National Oceanographic Center, Southampton SO14 3ZH, UK
[3]Institute of Marine Sciences, National Research Council (ISMAR-CNR), Venice, Italy
[4]Nansen Environmental and Remote Sensing Center, Bergen, Norway
[5]GEOMAR Helmholtz Centre for Ocean Research Kiel, Kiel, Germany
[6]Thales Alenia Space, Toulouse, France
[7]OceanDataLab, Locmaria Plouzané, France
[8] LEGOS, Université de Toulouse, CNES, CNRS, IRD, Toulouse, France
[9]University College, Dublin, Ireland
[10]CNES, Toulouse, France
[11]Division of Geological and Planetary Sciences, California Institute of Technology, Pasadena
[12]Earth Sciences Division, Jet Propulsion Laboratory, California Institute of Technology, Pasadena, California
[13]Environmental Hydraulics Institute "IH Cantabria" Universidad de Cantabria, Santander, Spain
[14]SCALIAN ALYOTECH, Rennes, France
[15]CLS, Toulouse, France
[16]Utrecht University, The Netherlands

*Correspondence to:* Fabrice Ardhuin (ardhuin@ifremer.fr)

**Abstract.** We propose a satellite mission that uses a near-nadir Ka-band Doppler radar to measure surface currents, ice drift and ocean waves at spatial scales of 40 km and more, with snapshots at least every day for latitudes 75 to 82, and every few days otherwise. The use of incidence angles at 6 and 12 degrees allows a measurement of the directional wave spectrum which yields accurate corrections of the wave-induced bias in the current measurements. The instrument principle, algorithm for current vector retrieval and mission performance are presented here. The proposed instrument can reveal features on tropical ocean and marginal ice zone dynamics that are inaccessible to other measurement systems, as well as a global monitoring of the ocean mesoscale that surpasses the capability of today's nadir altimeters. Measuring ocean wave properties facilitates many applications, from wave-current interactions and air-sea fluxes to the transport and convergence of marine plastic debris and assessment of marine and coastal hazards.

## 1 Introduction

Because the ocean surface is the interface between ocean, atmosphere and land, surface currents play an important role in defining the fluxes of heat, momentum, carbon, water, etc. between Earth System components. The ocean surface velocity combines surface currents (mainly driven by winds, density gradients, and tides), with a wave-induced drift, known as Stokes drift. This total mass movement transports surface heat, salt and everything that is in the upper ocean, natural or man-made, including marine plastic debris (van Sebille et al., 2015). While vastly improving over the satellite era, there are still important gaps in our knowledge of ocean currents and waves. Satellite altimeters have been around for over 25 years, revealing mesoscale ocean dynamics and providing a global view of the wind-generated waves.

Still, the along-track sea level anomaly misses most of the the multi-scale motions of surface ocean, because of a limited resolution (e.g. Fu and Ubelmann, 2014) and because a large component of surface currents is not in geostrophic balance. That second aspect is particularly relevant near the equator (Cravatte et al., 2016), in strong western boundary currents (Rouault et al., 2010; Rio et al., 2014), and everywhere due to near-inertial motions (Kim and Kosro, 2013; Poulain et al., 2016; Elipot et al., 2016). Around the edge of the sea ice, both altimeter and SAR processing difficulties and the small scales of currents conjure in making it a blind spot in today's observation systems (Korosov and Rampal, 2017).

Other available measuring systems are very local, such as HF radars, or global with a sparse coverage, such as drifters (e.g. Elipot et al., 2016). As shown on figure 1, a single polar orbiting satellite with a swath width of 270 km could extend the capability of existing systems for monitoring ocean surface velocities, in particular for wavelengths between 60 and 1000 km and periods ranging from 3 days to 30 days. Because larger scales move slower, the coarser time resolution at the equator also yields a coarser spatial resolution.

As detailed below, the Sea Surface KInematics Multiscale monitoring (SKIM) mission, propose to use map surface waves and currents with 6-km footprints resolved at 4 m resolution in range. These footprints are distributed across a 270 km wide swath, but do not cover the entire swath, leaving a gap between the features smaller than 6 km resolved with a footprint, and the features larger than 20 km fully mapped across the swath. As the ocean is viewed in less than 1 minute during a single pass, the observed scene is basically a snapshot in which many ocean processes are aliased. Only those current features that vary on time scales of several days, or that have a constant phase and amplitude such as tides, can be measured without ambiguity. Evidence from High Frequency radars in coastal areas suggests that even near-inertial motions are coherent over time scales as large as 6 days at mid-latitudes (Kim and Kosro, 2013). Hence measured currents, even if every 3 days only, can provide useful constraints on the ocean circulation.

Measurements of ocean surface currents from remote sensing platforms have used a wide range of techniques. The most widely used at large scales include satellite altimeters, possibly combined with scatterometer wind and in situ drifters (e.g. Bonjean and Lagerloef, 2002; Sudre et al., 2013; Rio et al., 2014). Other techniques such as image processing of optical or Synthetic Aperture Radar (SAR) imagery have been demonstrated in many regions (see Isern-Fontanet et al., 2017, for a

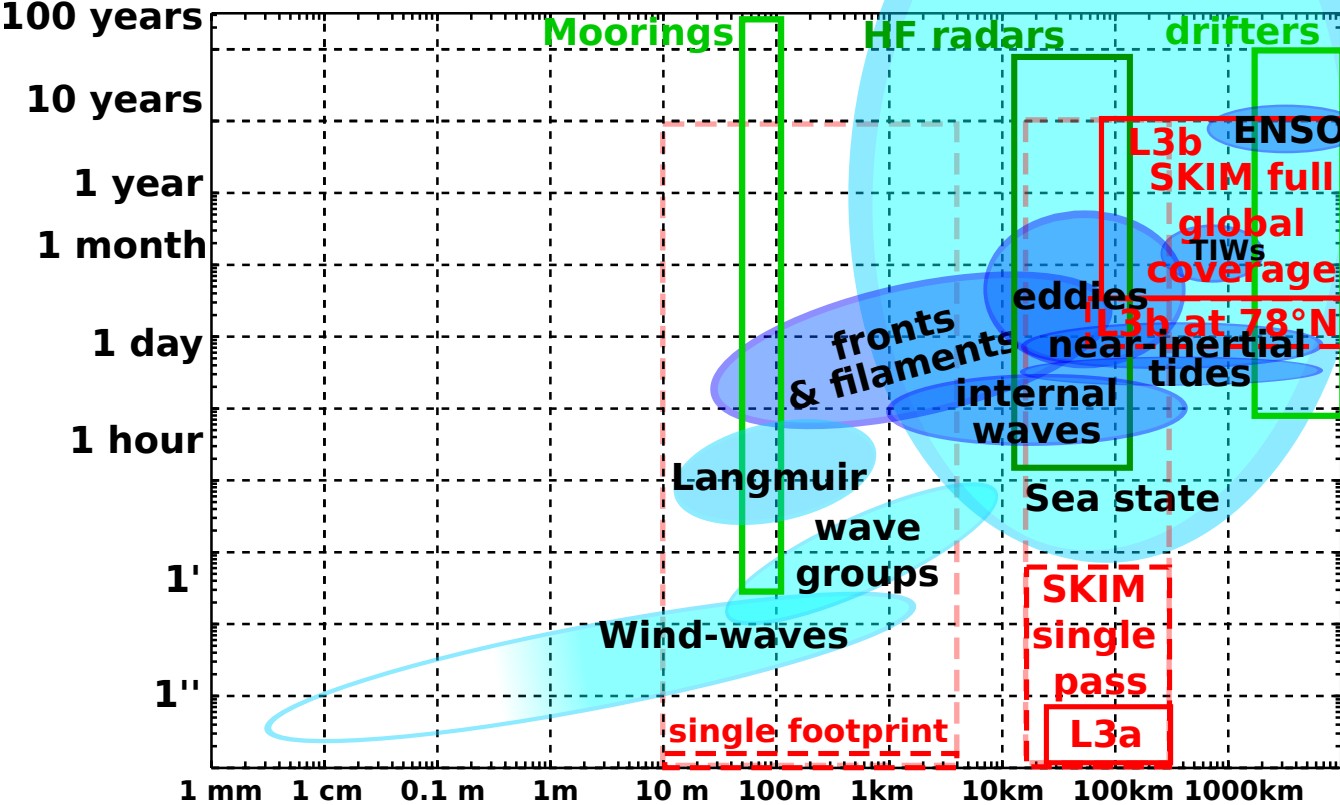

**Figure 1.** Typical periods and wavelengths of processes that contribute to the surface velocity. Ocean circulation processes are in dark blue, wave-related processes are in light blue. Scales resolved by existing measuring systems appear in green, and our proposition of a Sea surface KInematics Multiscale (SKIM) satellite mission in red, with different scales resolved in a single pass level 3a product (L3a), or the full time history of the measurements gridded as a level 3b product (L3b). Note that a limited coverage in space or time leads to aliasing of the unresolved scales that are in the pink boxes (e.g. Stammer et al., 2000; Gille and Hughes, 2001). Due to the polar orbit, the resolved periods varies from 12 days at the equator (0°) to 1 day at 78° latitude.

review). Direct measurements of the ocean surface current vector using Doppler techniques is now widely used in land-based radar systems, with operational use for current mapping in the high-frequency band, ranging from 3 to 30 MHz (Barrick, 1972). Interesting results have also been reported with land-based microwave radars (Forget et al., 2006, 2016), but the measured radial velocity is not fully understood.

5    Airborne and space-based measurements of surface velocity have been performed with across-track interferometric (ATI) SARs using two antennas (Goldstein and Zebker, 1987). This has been generalized to squinted ATI SARs to provide the two components of the current vector (Buck, 2005; Wollstadt et al., 2016). More recently, Chapron et al. (2005) have shown the potential of using the Doppler centroid of ocean backscatter received by a single antenna. Although this measurement is more noisy than ATI, resulting in an effective coarser resolution, the velocity given by the Doppler centroid is equivalent to an ATI

measurement (Romeiser et al., 2014). Hence, the Doppler centroid method is a cost-effective solution for deriving current information from existing satellite missions such as Envisat and the Sentinel 1 constellation. This has already led to scientific application on the monitoring of intense currents (Rouault et al., 2010).

This demonstration of Doppler oceanography from space, using measurements of opportunity, has led us to propose a specially built Doppler radar altimeter that uses nadir and off-nadir beams in Ka-band. SKIM is designed to measure both the horizontal surface velocity vector $(U,V)$, i.e. surface current or ice drift, and the directional wave spectrum $E(k,\theta)$ where $k$ is the wavenumber and $\theta$ is the azimuth of wave propagation. Wave spectra are used to correct for a wave-induced bias in the Doppler velocity. The purpose of the present paper is to describe the measurement principle and the expected instrument performance based on a preliminary analysis.

Doppler measurements start from a line of sight velocity $U_{\mathrm{LOS}}$ which contains a very large non-geophysical component $U_{\mathrm{NG}}$ due to the relative motion of the spacecraft relative to the solid Earth. The anomaly relative to $U_{\mathrm{NG}}$ can be interpreted as a horizontal geophysical Doppler contribution,

$$U_{\mathrm{GD}} = (U_{\mathrm{LOS}} - U_{\mathrm{NG}})/\sin(\theta_i) \tag{1}$$

where $\theta_i$ is the local incidence angle. The geometry of the measurement is illustrated in Figure 2.

Common to ATI and Doppler centroid techniques, is the contribution of orbital velocity of wind-waves to the geophysical velocity $U_{\mathrm{GD}}$, in the form of a wave bias $U_{\mathrm{WB}}$ (Chapron et al., 2005; Mouche et al., 2008; Martin et al., 2016), so that the radial current (projected on the mean sea surface in the azimuth of radar look) is

$$U_R = (U_{\mathrm{GD}} - U_{\mathrm{WB}}). \tag{2}$$

$U_R$ is the radial component of the Lagrangian mean velocity vector $\mathbf{U} = (U,V)$, defined from the average drift velocity of water parcels. This Lagrangian mean drift is $\mathbf{U} = \mathbf{U}_E + \mathbf{U}_S$, the sum of a quasi-Eulerian current (Jenkins, 1989) $\mathbf{U}_E$ and a Stokes drift $\mathbf{U}_S$ (Stokes, 1849). $\mathbf{U}_S$ is the surface drift vector due to waves, that arises from a correlation between the displacement and gradients of the velocity field: forward particle velocity at a crest is faster than the backward velocity at a trough. The Sto$U_S$ at the sea surface is of the order of 1.0 to 1.8% of the wind speed, typically larger than the local wind-induced quasi-Eulerian current known as the Ekman current, unless a strong stratification is present (Ardhuin et al., 2009).

Previous applications have used the radial wind speed $U_{10,R}$ projected in the range direction as a proxy for estimating $U_{\mathrm{WB}}$. As we review in section 2, this wind speed proxy is not sufficient for obtaining accurate instantaneous current velocities. We therefore propose in section 3 an algorithm for estimating $U_{\mathrm{WB}}$ within 10 to 20%, based on the measurement of waves with a rotating radar system. This technique forms the conceptual basis for SKIM. Its expected overall performance and effective resolution is described in section 4. A summary and perspectives on applications and improvement in the processing follow in section 5. The present paper focuses on currents, and a detailed description of wave measuring capabilities with SKIM will be given elsewhere.

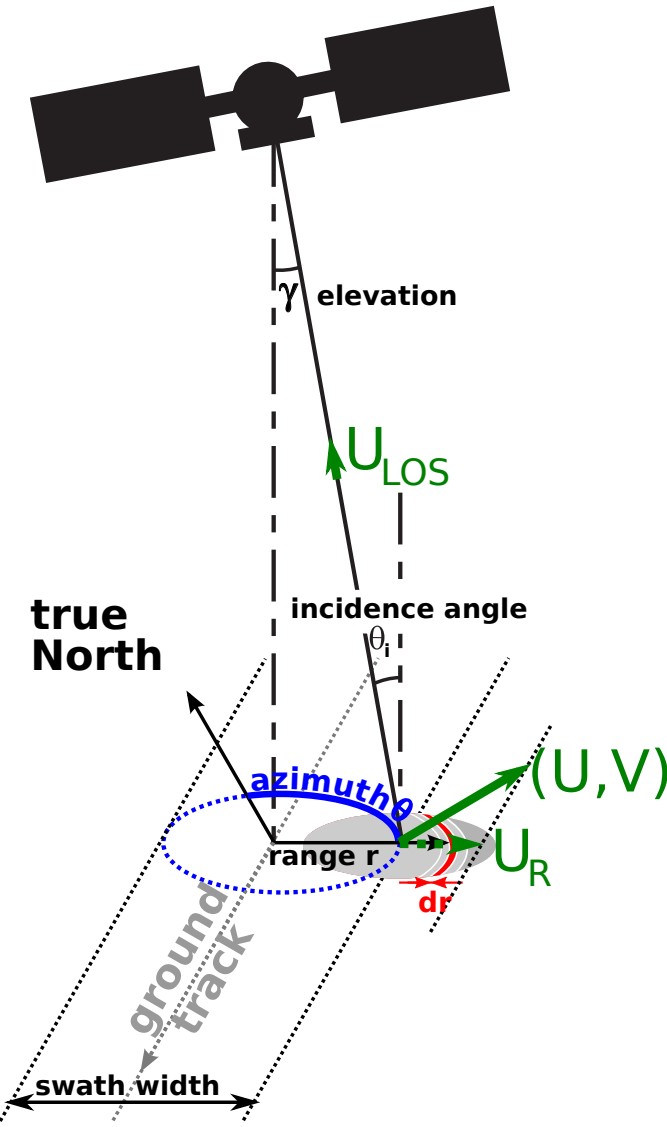

**Figure 2.** Geometry of measurement from a radar with a local incidence angle $\theta_i$, looking towards azimuth $\theta$. For simplicity of the schematic, we have taken $U_{\mathrm{WB}} = 0$ and $U_{\mathrm{NG}} = 0$, so that the line of sight velocity is simply $U_{\mathrm{LOS}} = (U \sin\theta + V \cos\theta) \sin\theta_i$. Note that the diameter of the footprint (6 km) is exaggerated compared to the swath width (270 km) for readability. The small difference between elevation $\gamma$ and incidence angle $\theta_i$ is due to the Earth curvature.

## 2 Importance of mean slope speed or Stokes drift

### 2.1 Expected and observed dependence of $U_{\mathrm{WB}}$

Because the velocity or phase shift recorded by a radar corresponds to the velocity weighted by the back-scattered power, the wave-induced bias $U_{\mathrm{WB}}$ is related to the mean slope velocity vector, $\mathbf{msv} = (<\partial^2\zeta/\partial x\partial t>, <\partial^2\zeta/\partial y\partial t>)$, due to the correlation between the normalized radar cross section (NRCS or $\sigma_0$) and the surface slope (e.g. Nouguier et al., 2018). For linear waves, $\mathbf{msv}$ is equal to twice the surface Stokes drift vector $(U_S, V_S)$.

In practice $U_{\mathrm{WB}}$ is very close to a gain factor $G$ multiplied by $U_{S,R}$, the surface Stokes drift projected on the range direction (Chapron et al., 2005) with an additional correction proportional to the Stokes drift in the azimuthal direction (Nouguier et al., 2018). $G$ is a function of radar frequency, incidence angle, and sea state. Figure 3 shows the expected dependence of $G$ on the incidence angle for average wind speeds and a fully developed sea state, using a physical optics model or Kirchoff approximation in the upwind direction (e.g. Nouguier et al., 2018). A typical order of magnitude in Ka-band for incidence angles less than 15 degrees, is $G \simeq 25$, which is similar to values in C-band at higher incidence angles.

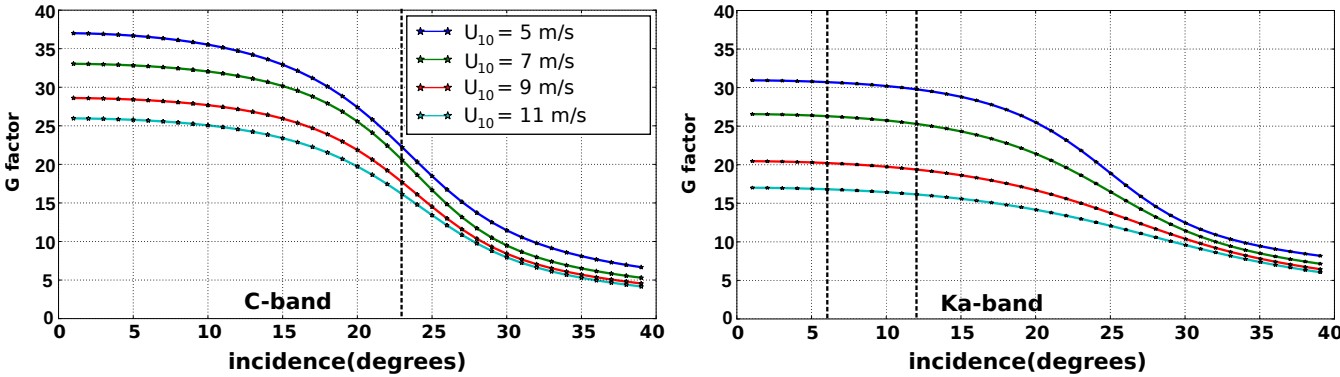

**Figure 3.** $G$ factor in the upwind looking direction estimated using a Kirchoff Approximation, for a wave spectrum given by Elfouhaily et al. (1997), representing a fully developed sea state for wind speeds $U_{10}$ ranging from 5 to 11 m/s. left: C band, appropriate for Envisat and Sentinel 1, right: Ka band for SKIM. The vertical dashed lines show the incidence angles of the instruments.

This dependency of $U_{\mathrm{WB}}$ on the radial Stokes drift $U_{S,R}$ and incidence angle $\theta_i$, as well as the order of magnitude of $G$ are confirmed by the analysis of platform-based measurements by Yurovsky et al. (2018), and airborne measurements from the AirSWOT instrument (e.g. Nouguier et al., 2018).

### 2.2 Estimation of $U_{\mathrm{WB}}$

The surface Stokes drift vector $\mathbf{U}_S = (U_S, V_S)$ can be estimated from the directional wave spectrum, assuming linear wave theory (Kenyon, 1969). The wave spectrum $E(k, \theta)$ represents the distribution of the surface elevation variance across wavenum-

bers $k$ and azimuthal wave propagation directions $\theta$. For waves in deep water this is

$$(U_S, V_S) = 2\sqrt{g} \int\limits_{0}^{2\pi} \int\limits_{0}^{\infty} (\sin\theta, \cos\theta) k^{1.5} E(k,\theta) \mathrm{d}k \mathrm{d}\theta. \tag{3}$$

This integral can be estimated from the first moments $a_1$ and $b_1$ measured by directional wave buoys from the co-spectra of vertical and horizontal accelerations (e.g. Kuik et al., 1988).

The projected Stokes drift $U_{S,R}$ is correlated with the wind speed in the radial direction $U_{10,R}$. Hence, the approximation of $U_{\mathrm{WB}}$ as a function of $U_{10,R}$ is a logical first step proposed by Chapron et al. (2005) and Mouche et al. (2008), and used by Rouault et al. (2010) to retrieve surface currents.

However, for a given wind speed the sea state introduces a typical variation of $U_{S,R}$ that has a standard deviation of 40%. Further, the distribution of $U_{S,R}$ as a function of $U_{10,R}$ can change significantly from one region of the ocean to another. These

properties are illustrated in figure 4 with data for the years 2011 to 2015, from the North-East Pacific station PAPA, in deep water (Thomson et al., 2013), and a North-East Atlantic coastal buoy Pierre Noires, in 60 m depth (Ardhuin et al., 2009). In both cases the wind speed is taken from operational ECMWF analyses. Directional wave moments were downloaded from CDIP and CEREMA. The Stokes drift was integrated over the frequency range of the Datawell Waverider buoy, from 0.025 to 0.58 Hz.

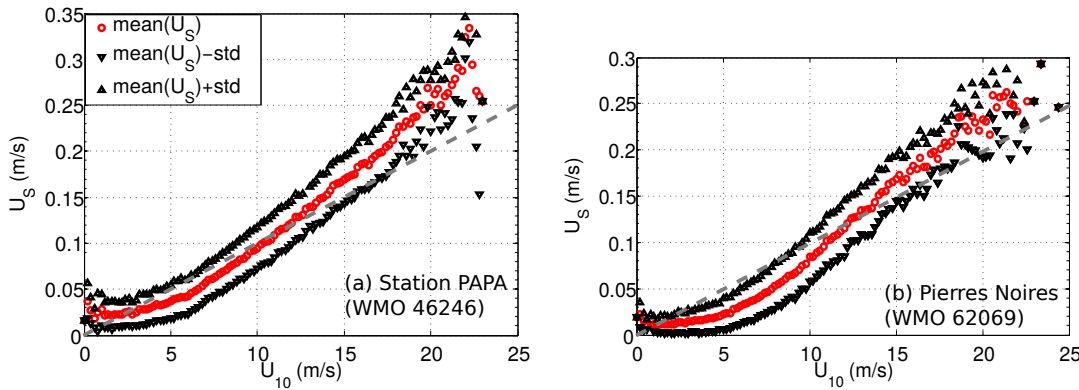

**Figure 4.** Example of mean value (in red) of the Stokes drift vector norm $U_S = |(U_S, V_S)|$ as a function of wind speed for two locations: station PAPA in the North-East Pacific, and buoy 62069 off the French Atlantic coast. The black symbols show the mean plus or minus one standard deviation for each wind speed. The dashed grey line is $U_S = 0.01 U_{10}$. This estimation covers only frequencies up to 0.58 Hz.

Using the order of magnitude $U_S \approx 0.01 U_{10}$, the sea-state variation means that a wind-only proxy for $U_{\mathrm{WB}}$ gives a root mean square (rms) error on the current of the order of $40\% \times G \times 0.01 = 10\%$ of the wind vector. With a median wind speed of 7 m/s, this is a 70 cm/s error in the wind direction for C-band at $23°$ of incidence, or Ka-band at $12°$. Such a high value is not acceptable for a single satellite pass, but these errors cancel out when the Doppler velocity is averaged over many satellite passes, 10 or more, as done by Collard et al. (2008) and Rouault et al. (2010). Even at the higher incidence angles of $58°$

proposed by Bourassa et al. (2016), for which we expect $G \simeq 7$ in Ka-band, the wave bias is reduced by a factor 4, but the rms

error on $U_{\mathrm{WB}}$ is still significant at 20 cm/s, even if there is no error on the wind. Larger incidence angles also suffer from lower backscatter levels and thus a larger instrumental error in the raw line of sight velocity $U_{\mathrm{LOS}}$.

A possible intermediate approach is to use a numerical wave model to estimate $U_{S,R}$, with typical errors ranging from 15 to 20% in open ocean and deep water conditions according to Rascle and Ardhuin (2013). Yet, recent investigations by Ardhuin
et al. (2017b) on the impact of ocean currents on small scale sea state variations suggest that it may be difficult to separate the gradients in wave bias from the surface current at scales under 100 km.

Another more radical approach is to measure the sea state properties necessary for the evaluation of $U_{S,R}$, in addition to the Doppler velocity $U_{\mathrm{LOS}}$. In general $U_{S,R}$ can be estimated from the directional wave spectrum. The details of this estimation with a rotating wave Doppler spectrometer, combining the ideas of Jackson et al. (1985) and Caudal et al. (2014), is presented
in Appendix A. An overall accuracy of 10% for $U_{S,R}$ is expected from our preliminary algorithm.

## 3 Restitution of the total surface velocity

The algorithm proposed to retrieve the field of surface velocity vectors and wave spectra is summarized in Figure 5. The elementary measured quantities are the power $P$ and velocity $U_{\mathrm{LOS}}$ as a function of the range $r$ within each footprint of diameter 6 km, with a resolution $dr$ that is determined by the 200 MHz radar bandwidth giving 0.75 m along the line of sight,
and less than 4 m projected on the horizontal for $\theta_i = 12°$. The range-averaged line of sight velocity is also given by the phase difference between pairs of pulses (Zrnic, 1977). This estimate requires a strong correlation between conscutive echoes in spite of rapid motion of the footprint, which calls for a relatively high pulse repetition frequency, and thus the averaging over many pulses to reduce the random error on small phase shifts. We also note that the horizontal current contribution to $U_{\mathrm{LOS}}$ occurs through the apparent vertical motion of the surface as waves are advected by the current (Nouguier et al., 2018). As a result,
for depth-varying currents, the measured current corresponds to the advection velocity (Kirby and Chen, 1989) for waves contributing to the mean slope velocity. In summary, the basic measurement are highly resolved in range but averaged over the footprint diameter in the perpendicular (azimuthal) direction. This averaging is the basic principle of the wave spectrometer laid out by Jackson et al. (1985). Namely, only the waves aligned with the line of sight produce a modulation of the signal in the range direction (see also Nouguier et al., 2018, eq. 46).
Several effects introduce measurement errors. We have particularly investigated the following three terms in the error budget for the level 2 data (radial current velocity $U_R$),

- $\mathrm{err}_{\mathrm{DC}}$: The Doppler centroid estimation error is a function of the strength of the radar backscatter, hence of the incidence angle, radar transmitted power, altitude and averaging. Using broad margins (e.g. using only half of the rated Ka-band power) this was estimated to be under 10 cm/s at $\theta_i = 12°$ and in the absence of ice (The SKIM Team, 2017). This error
is a well known function of the azimuth relative to the satellite ground track.

- $\mathrm{err}_{\mathrm{PA}}$: A 5 cm/s second error corresponds to a Doppler shift of 56 Hz that can be obtained by an error in elevation of $8 \times 10^{-5}$ degree (see Appendix B). However, such an error is easily detected thanks to the rotating beam. This is because

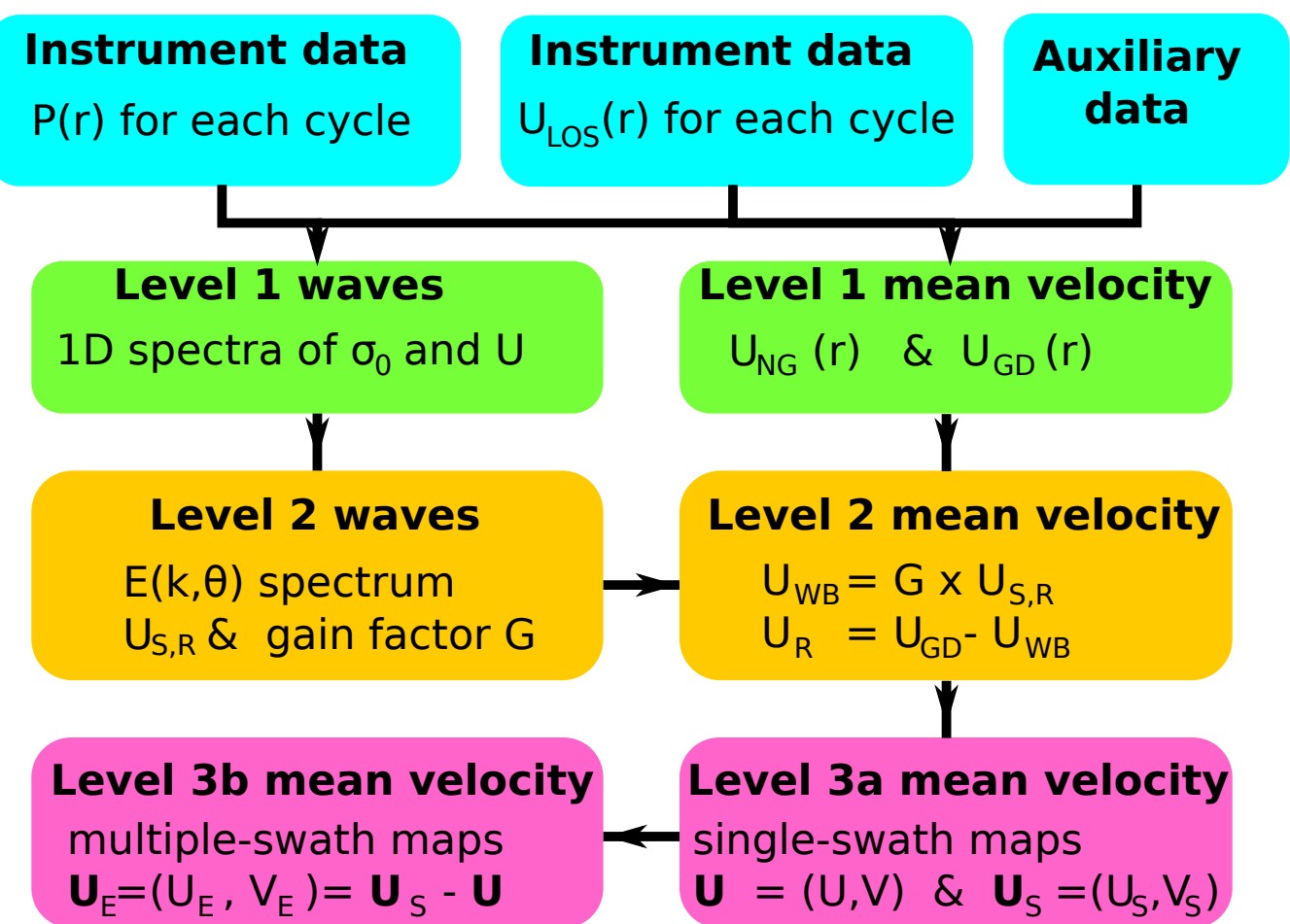

**Figure 5.** Logical tree going from Level 0 raw data to Level 3 gridded fields of surface velocity and wave parameters. For gridding with multiple satellite passes, we propose to use the quasi-Eulerian surface velocity defined as $\mathbf{U}_E = \mathbf{U} - \mathbf{U}_S$ (Jenkins, 1989).

any small mispointing that varies slowly in time produces a clearly identifiable pattern as a function of azimuth. This is detailed in Appendix B. As a result, with a realistic attitude stability better than $2.7 \times 10^{-4}$ degree over 20 s, we expect a rms contribution to the current error of a few cm/s. Our worst case scenario with random jumps of the attitude gave errors of 3 cm/s. This error will be neglected in the following sections.

5    – $\text{err}_{WB}$: The wave bias error is explained and justified in Appendix A. We have performed a detailed analysis on the error $\text{err}(U_{S,R})$ in the estimation of the radial Stokes drift $U_{S,R}$ over each footprint, which contributes to $\text{err}_{WB}$ amplified by the $G$ factor. Errors in the estimation of the $G$ factor are not so easy to model but, for a given mean square slope, $G$ is expected to have a weak dependence on sea state properties and it is related to the ratio of the Doppler and $\sigma_0$ spectra. We have therefore assumed that errors on the estimation of $G$ should not cause an error larger than the error due to

10    uncertainties in $U_{S,R}$. Hence we used $\text{err}_{WB} = 2G\,\text{err}(U_{S,R})$

We have not considered the particular cases of extremely low backscatter, for wind speeds under 2 m/s, in which the three errors can be correlated, and we have assumed that these 3 error sources are uncorrelated.

## 4  Overall performance and effective resolution

### 4.1  From radial components to gridded vector fields

5     Here we show results corresponding to one particular set-up of the SKIM radar, which is called 'SKIM2' (see The SKIM Team, 2017, for details). This configuration uses 8 beams, with one beam at nadir ($\theta_i = 0$), two beams at 6° and 5 beams at 12°. These beams rotate at 3.14 rotations per minute (one turn in 17.5 s) thanks to the rotation of a plate carrying feed horns arranged as shown in Figure 6.a.

    The horns are placed around the focal point of a parabolic reflector, similar to the wave scatterometer SWIM of the China-

10 France Ocean Satellite (CFOSAT) mission (Hauser et al., 2017). The main differences between SWIM and SKIM are the radar frequency (Ka instead of Ku band, giving smaller footprints), and the Doppler capability of SKIM. Using incidence angles up to 12° and altitude of 695 km gives swath width of 270 km as shown in Fig. 6.b.

**Table 1.** Summary of expected root mean square errors for Level 2 radial velocity (for $\theta_i = 12°$) and along-track Level 3a (single swath snapshot) or zonal Level 3b (multi-swath time-evolving field) velocity component, based on the preliminary algorithms in the case of the SKIM2 configuration (open burst, 8 beams).

| Region | Equator | Fram (open water) | Fram (ice) | Gulf Stream | Oregon coast |
|---|---|---|---|---|---|
| $err_{DC}$ | < 0.1 m/s | < 0.1 m/s | < 0.1 m/s | < 0.01 m/s | < 0.01 m/s |
| $err_{PA}$ | < 0.03 m/s | < 0.03 m/s | < 0.03 m/s | < 0.03 m/s | < 0.03 m/s |
| $err_{WB}$ | 0.05 m/s | 0.08 m/s | 0.02 m/s | 0.15 m/s | 0.13 m/s |
| L3a, $L_e$ | 89 km | 59 km | TBD | 65 km | 90 km |
| L3a error | 0.03 m/s | 0.11 m/s | TBD | < 0.09 m/s | 0.04 m/s |
| L3b, $L_e$ | 290 km | 62 km | TBD | 71 km | 95 km |
| L3b error | 0.14 m/s | 0.12 m/s | TBD | 0.23 m/s | 0.09 m/s |

    For each beam, this gives 60 measurement cycles of 1024 pulses each with a 36.6 ms duration for each cycle. The radar pulse repetition frequency is 32 kHz. The line-of-sight velocity is determined from the phase shift between consecutive pulses.

15 These parameters define the instrument error $err_{DC}$, as listed in table 1.

    The other important source of error, caused by inaccuracies in the wave bias correction is a function of the beam geometry but also of the strength of gradients of the Stokes drift, which are mostly caused by current gradients (Ardhuin et al., 2017b). This wave bias error is estimated following the method laid out in Appendix A.

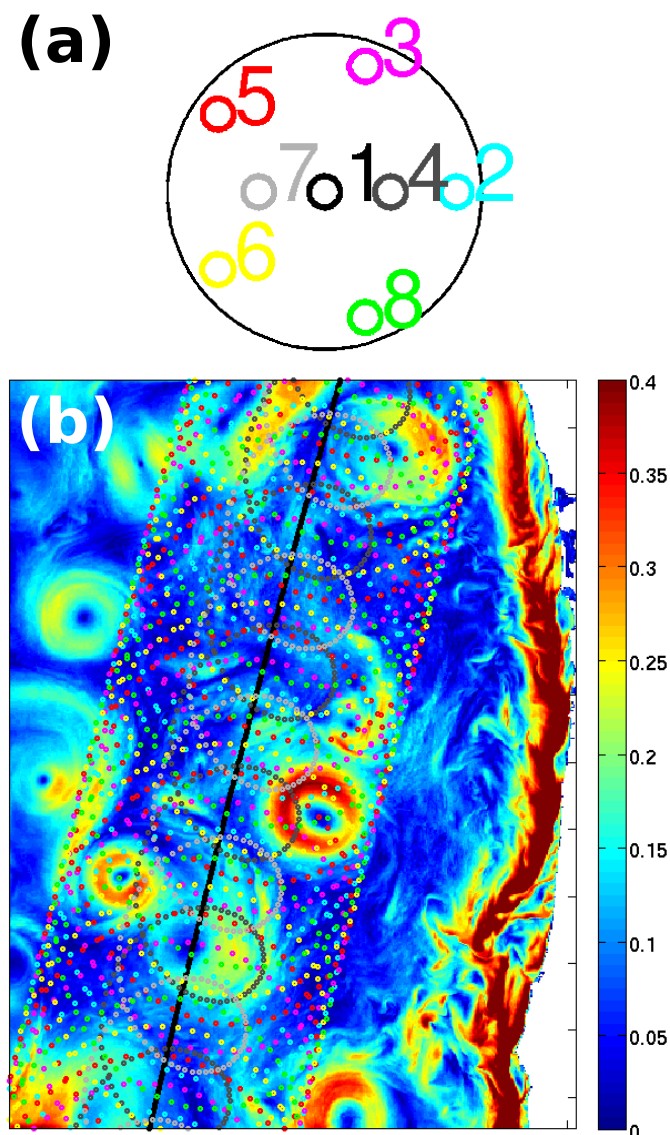

**Figure 6.** (a) Pattern of beams (1 is nadir, 4 and 7 are at 6°, the others at 12° incidence). The different colors help associating the footprint patterns in (b) with each beam. Background colors in (b) represent simulated currents velocities off the Oregon coast (in m/s), courtesy of Y. Chao, previously used by Fu and Ubelmann (2014) for the evaluation of the SWOT mission performance.

Finally, the last important source of error we have investigated is the mapping error, going from Level 2 data at each footprint to Level 3 data on a regular grid. This mapping error is similar to what happens with HF radars (e.g. Lipa and Barrick, 1983; Kim et al., 2008). In particular, SKIM only measures radial components so that on the edges of the swath only the cross-track component is measured, and in the center there are only measurements of the along-track component, as shown in figure 7.

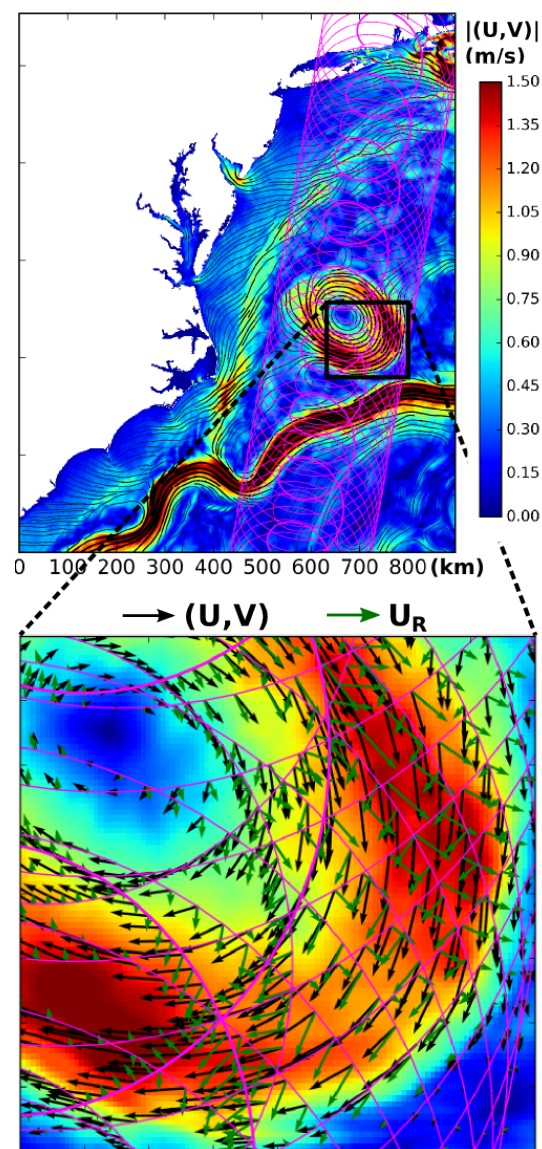

**Figure 7.** Illustration of current components $U$ and $V$ and radial component $U_R$ in the case of the Gulf Stream.

The use of the two incidence angles, 6 and $12°$, allows to fill the swath and obtain cross-track measurements closer to the center of the swath. We can also use the nadir altimeter beam to obtain cross-track geostrophic velocities. An optimal interpolation method specially designed to include covariances between the two current components has been adapted to also include this additional nadir data (The SKIM Team, 2017).

5      The combination of these three errors gives the total error that must be compared to the magnitude of the current. We have therefore defined an effective resolution wavelength $L_e$ as the scale above which the total error is larger than the signal, as

shown in Figure 8 for the case of the Gulf Stream. The overall error depends on many factors related to the patterns in ocean

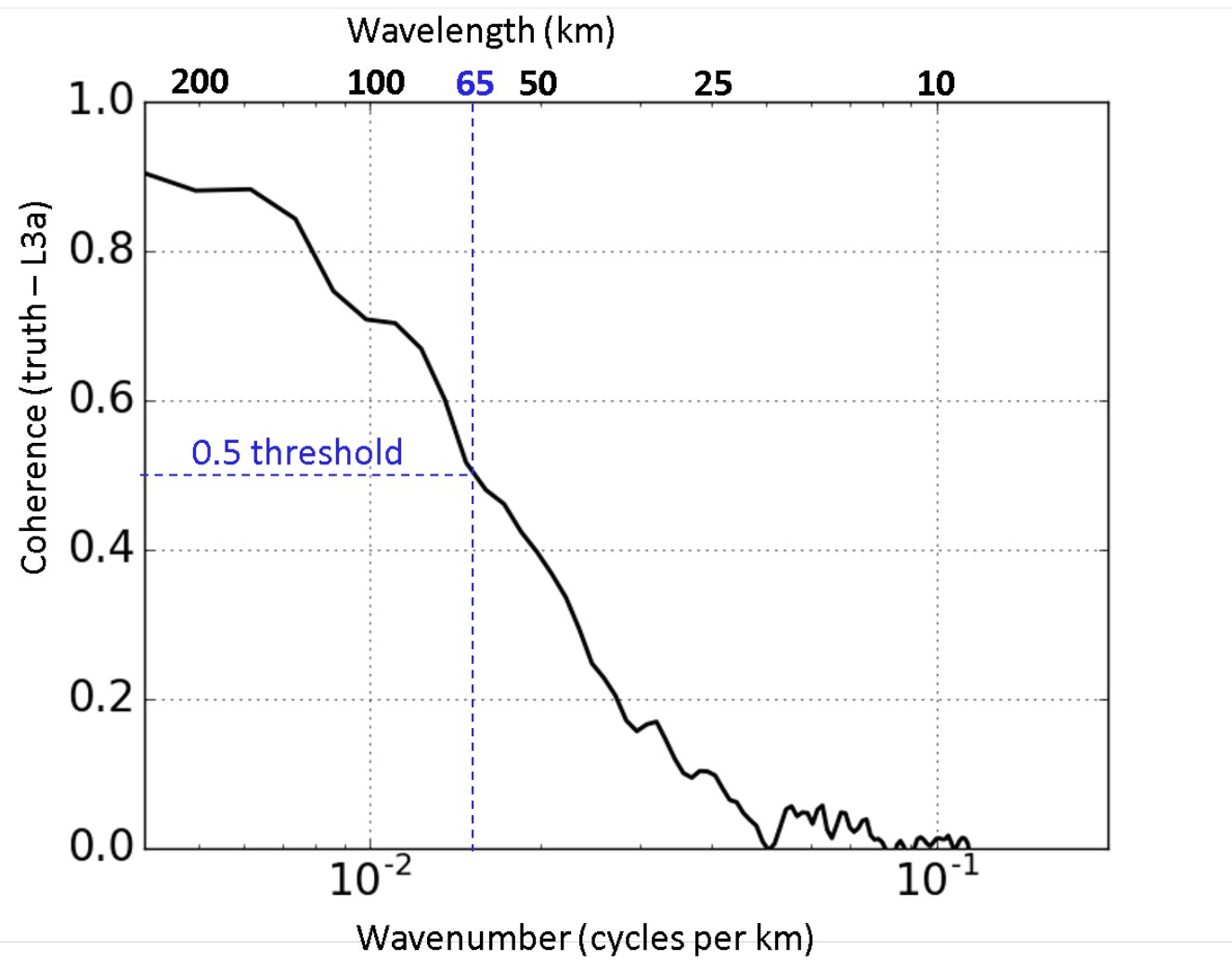

**Figure 8.** Average coherence of simulated SKIM Level 3a current with the "truth" provided by MITgcm simulations, in the case of the Gulf Stream, for October 2011.

currents and the instrument parameters. We have estimated all errors in particular ocean conditions, using state of the art models for the ocean circulation (e.g. Rocha et al., 2016; Gula et al., 2015) and associated ocean waves (Roland and Ardhuin, 2014; Ardhuin et al., 2017b). Model simulations were performed at resolutions on the order of 1.5 km for a set of regions for which we expect SKIM to have a strong contribution, resolving processes that are not accessible with today's observing

systems. These include an Arctic region with strong currents (Fram Strait), an equatorial region (in the Atlantic around 23°W), a western boundary current (the Gulf Stream) and a coastal region (Oregon).

The root mean square error on current components and the resulting effective resolution are summarized in Table 1, but they are better understood by comparing maps of currents from the simulated SKIM processing to the input modeled currents. Several examples are given by The SKIM Team (2017). It is also interesting to compare the results of different observing systems. An important outcome of the SKIM simulations is that a wide swath is necessary to obtain the shorter revisit time needed to monitor the smaller ocean structures that evolve more rapidly.

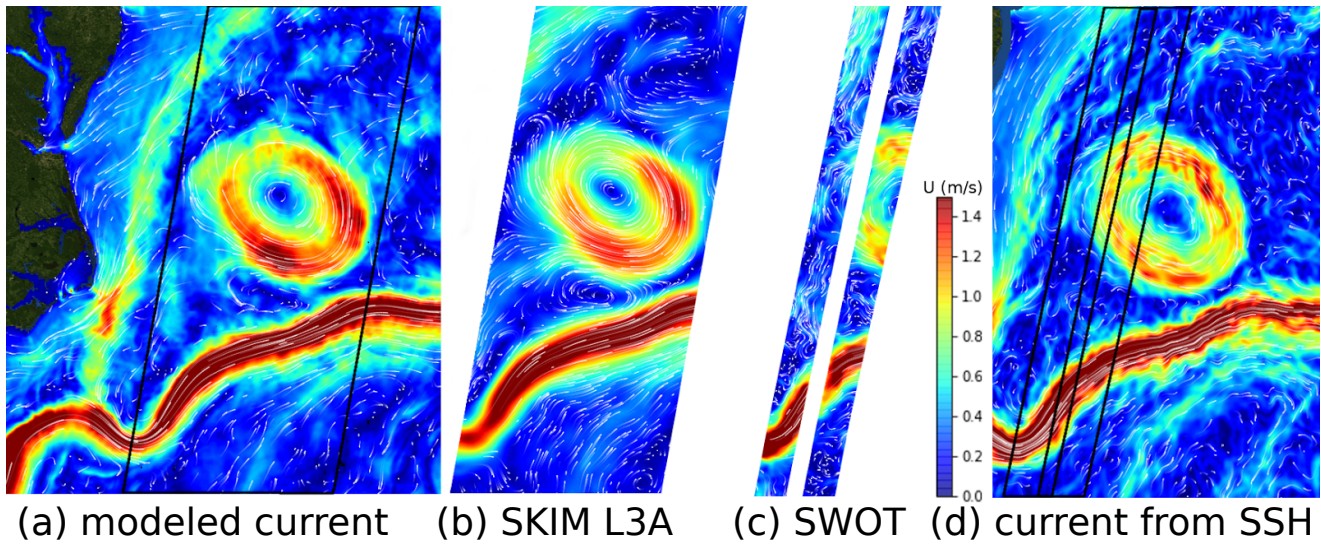

**Figure 9.** Simulated ocean currents over the Gulf Stream, and associated SKIM and SWOT simulated observations for a single satellite pass.

Interferometric SAR technology will be used to produce a 120 km wide swath for the Surface Water Ocean Topography (SWOT) mission (see Figure 9). The narrower swath gives a larger revisit time at mid-latitudes of 10 days with SWOT instead 4 with SKIM. As a result, only larger scale motions can be monitored with SWOT, with an effective wavelength $L_e = 115$ km in the Gulf Stream region, instead of 71 km with SKIM. The Doppler scatterometer mission proposed by Bourassa et al. (2016) has a much wider swath, about 1800 km and is designed to measure wind and current vectors. Because it measures at larger incidence angles for which signals are weaker, an accurate current estimate requires averaging over several passes, thereby reducing the effective temporal resolution. As a result, this is a great instrument for vector wind measurements but it is not clear if it would perform better than SKIM for current measurements.

In the case of SWOT, the interpretation of the sea surface height (SSH) in terms of current relies on the geostrophic equilibrium. Unbalanced motions, such as internal waves, also contribute to the SSH. As a result this simple interpretation of the SSH contains small scale noise associated to ageostrophic motions, shown in Figure 9.c,d. The separation of balanced and unbalanced motions is the topic of active research (e.g. Ponte et al., 2017). Further improvements in the restitution of temporal

evolution, and thus the reduction for $L_e$ for L3b products from both SWOT and SKIM, will benefit from dynamical methods which are under development (Ubelmann et al., 2016).

## 4.2    Challenges and opportunities over sea ice

With the expected widening of the Marginal Ice Zone (MIZ) in the Arctic (e.g. Aksenov et al., 2017), this expanding and important region of the world ocean will not be well monitored in terms of currents by existing and planned satellite missions.

Ice concentration is the only parameter that is well monitored near the ice edge, with difficulties in recovering ice thickness and ice drift (Korosov and Rampal, 2017). The very rich dynamics across the ice edge offer great opportunities for Doppler-based measurements. In particular, narrow ice jets and eddies are observed in satellite imagery (Johannessen et al., 1983) and reproduced in high resolution models (Horvat et al., 2016).

These features cannot be monitored by today's altimetry due to their small scale and the changes in waveform shapes from open water to ice. In the ice, the wave-induced bias becomes negligible as the wave amplitude is strongly attenuated. On the contrary, the instrument noise is expected to increase by about a factor 2.5 due to a generally weaker (8 dB) back-scatter over ice compared to open ocean at incidence angles under $12°$ (The SKIM Team, 2017). A detailed analysis of errors right at the ice edge requires to take into account the strong variation in backscatter in all terms of the error budget. This is beyond the scope of the present paper.

Also, it should be possible to measure waves in ice, without the SAR processing used by Ardhuin et al. (2017a) but using the Doppler spectrum and the modulation due to range bunching. Indeed, it is not clear how strong the tilt modulation is over the ice, but range bunching is maximum for a swell steepness $ka = \tan\theta_i$, which is 0.1 for $\theta_i = 6°$. A swell steepness of 0.025, as in the Marginal Ice Zone observations reported by Sutherland and Gascard (2016), still produces a 20% (0.9 dB) modulation of $\sigma_0$.

## 5    Preliminary study of surface current impact

In order to evaluate the contribution of a surface Doppler measurement in an ocean forecasting system, we have used the TOPAZ assimilation system, implemented in the Copernicus Arctic Marine Forecasting Center. This system uses a regional configuration of the HYCOM ocean model over the North Atlantic and Arctic Oceans - without tides - and assimilates different types of satellite and in situ observations with an Ensemble Kalman Filter, running 100 dynamical members.

Each ensemble member receives random perturbations of the ocean surface conditions, including non-divergent random winds with an amplitude of 2.5 m/s (see Xie et al., 2017, for more information about the reanalysis). We have used the simulated uncertainties of SKIM Level 2 surface velocities, following their description above, to produce a measure of the impact of assimilating SKIM surface currents in conjunction to all other observations on a typical weekly cycle of the TOPAZ reanalysis in May 2015, in a period of stable reanalysis operations following 24 years of data assimilation.

We measure the information content of the assimilated data using the Degrees of Freedom for Signal (DFS) (Cardinali et al., 2004). This DFS has a maximum value of 100 in the case of the EnKF used in TOPAZ. Target DFS values range from 0 to 10,

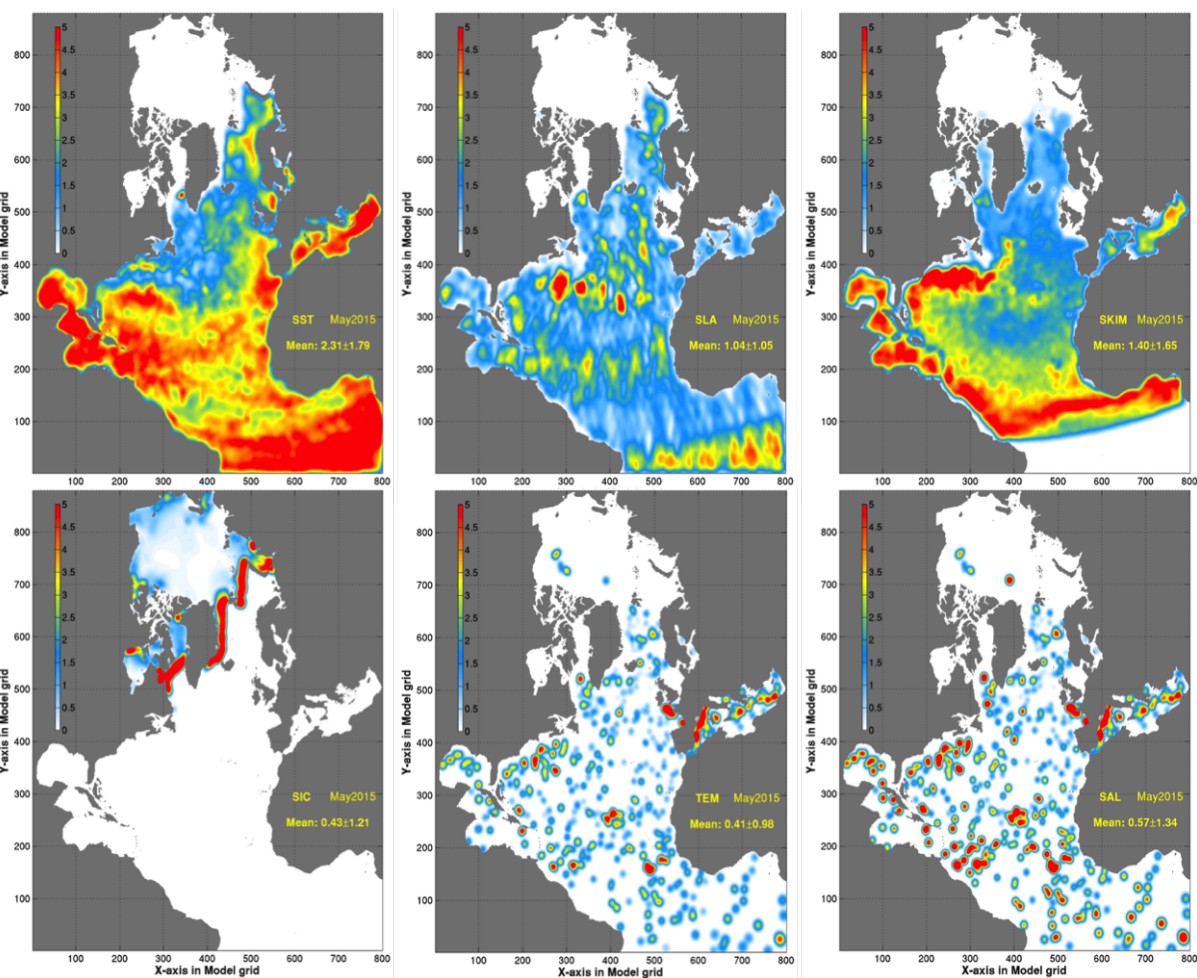

**Figure 10.** Degrees of Freedom for Signal of all assimilated observations during one week in May 2015. From Top line from left to right: OSTIA Sea Surface Temperature, CMEMS delayed-mode altimeter tracks, SKIM sea surface. Bottom line: OSI SAF Sea ice concentrations, in situ (incl. Argo) Temperature and Salinity profiles

above which there is a risk of "over-assimilation" (Sakov et al., 2012). The observation impact is calculated for each grid cell, using knowledge of the space and time location of the observations, their uncertainty estimates, but not the actual observed values. Since all observations are assimilated jointly, the impact of one observation type reduces that of the other types. The DFS values are dependent on the background and observation uncertainties specified in the TOPAZ system and are different in a different ocean data assimilation system.

Figure 10 exhibits the DFS values obtained by assimilating simulated SKIM surface currents together with other real measurements. The highest DFS appears in frontal regions like the Equatorial Counter-Current, the Gulf Stream and the Azores Current. The area near the Equator shows particularly high values as the impact of traditional altimeter data is limited by the

vanishing of the Coriolis force. The South Atlantic is artificially removed as the simulated SKIM data used here only covers the North Atlantic.

The DFS indicate that SKIM provides the second largest impact overall, and the largest information content in the Gulf Stream and Equatorial regions for ocean data assimilation. It may seem counter-intuitive that the impact of surface currents from SKIM exceeds the impact of depth-averaged currents as of traditional altimeters in the Gulf Stream. This could be a transient effect due to the first-time assimilation of SKIM: the ensemble variance of surface currents may reduce on the following assimilation cycles and the DFS reduce accordingly. Alternatively it could be due to our assumption of negligible representativity errors.

The results presented here only utilize the surface currents but not yet the surface waves nor the sea ice drift from SKIM. This observing system simulation experiment (OSSE) is highly simplified and does not resolve complex feedbacks of repeated data assimilation cycles. Still, this OSSE indicates that there is a scope for assimilation of sea surface currents in an operational forecasting system and that SKIM data should provide relevant information that is independent of existing ocean observations.

## 6  Conclusions and perspectives

Using nadir and near-nadir radar beams with Doppler measurements, the Sea surface KInematics Multiscale monitoring (SKIM) mission is designed to measure surface velocity vectors and ocean wave spectra. Measuring wave spectra down to wavelengths of 20 m and possibly less makes it possible to estimate the surface Stokes drift vector and correct for a strong wave-induced bias in the surface velocity vector, which is of the order of a gain coefficient $G$ times the surface Stokes drift.

The use of a rotating beam pattern is critical in reducing errors caused by knowledge uncertainties in the platform attitude, which is today the main source of error in the level 2 surface current derived from the Sentinel 1 SAR constellation. Here we presented a performance analysis using the orbit of Sentinel-1C (S1C), except for a 4° shift to the east. This is a sun-synchronous orbit, with a 98.2° inclination, altitude 693 km and 12-day repeat cycle. This geometry gives a swath width of 270 km and a relatively large signal to noise ratio thanks to the higher backscatter at these low incidence angles. The incidence angle of SKIM is limited to 12° by the choice of antenna technology that uses a rotating horn plate and fixed parabolic reflector. With this configuration, larger incidences lead to beam distortions. The 4° shift allows for a large overlap between SKIM and S1C on ascending tracks that could be useful for calibration purposes. Other choices in synergy with altimeters or radiometers could justify a higher altitude, allowing for a wider swath at the price of a lower signal to noise ratio for the radar detected power.

Compared to the Envisat C-band measurements at incidence angles of 30° used by Rouault et al. (2010), in which case $G \simeq 12$, the error on the wave bias is expected to be reduced by a factor 4 or more, allowing a single-pass estimation of the current components with an accuracy of the order of 0.1 m/s for a wavelength of about 60 km.

When the radial components are combined to produce maps of gridded vector velocity, the effective wavelength resolved, at which the signal is above the noise level, is of the order of 60 to 90 km for a single swath, depending on the pattern of currents.

Except for latitudes 78 to 83° where the revisit time is less than one day, the effective resolution is degraded when the time evolution of the currents is considered. At mid latitudes this gives $70 < L_e < 100$ km, due to the 3-day revisit time.

Further improvement on the accuracy and effective resolution may come from many improvements in radar settings (e.g. use of full power instead of 50%, evolution in amplifier technology) which could give larger transmitted power and reduce the

instrument noise $err_{DC}$. Another source of improvement will be the reduction of wave bias error $err_{WB}$, in which our estimation of the $G$ factor error may well be overestimated, and a combined analysis or assimilation of waves and currents could properly take into account the correlations of waves and currents and lead to more accurate current estimates. Hence the error level and resolution found in our simulations are probably conservative.

Our results clearly show that Doppler oceanography from space can be a very useful technique for monitoring space and

time scales of the ocean currents that are not well observed today. Future altimeter designs should probably consider adding off-nadir rotating beams for a more effective coverage of the ocean. In the present paper we have not discussed much the added benefits of ocean wave measurements with unprecedented spectral and spatial coverage. These will be discussed in other publications. We only point out here that the sea state variability at small scales is probably dominated by the effect of ocean currents (Ardhuin et al., 2017b). It is thus logical to measure waves and currents together, and possibly further use the measured

variability of sea state parameters to further constrain the magnitude of current gradients.

*Code and data availability.* Numerical model results presented in this article are available via ftp at the following address: ftp://ftp.ifremer.fr/ifremer/ww3/

## Appendix A: Estimation of wave-induced bias $U_{WB}$ from directional wave data

One important difficulty for the estimation of $U_{S,R}$ by projecting eq. (3) in direction $\theta$, is that the Stokes drift contains contributions from all directions $\theta'$ whereas the measurement on a single footprint only give contributions in the look direction $\theta$.

For each footprint in azimuth $\theta'$ we only have the contribution of the waves propagating in direction $\theta'$, which we define as

$$
\begin{aligned}
U_{S,1D}(\theta') \quad = \quad & 2\sqrt{g} \int\limits_{0}^{k_{max}} k^{1.5} E(k,\theta') \mathrm{d}k \\
& + F(k_{max},\theta') E(k_{max},\theta'),
\end{aligned}
\tag{A1}
$$

where $k_{max}$ is the wavenumber of the shortest resolved waves. Assuming that the spectrum of shorter waves rolls of like $k^3$ and neglecting non-linear effects gives $F(k_{max},\theta') = \sqrt{g}k_{max}^{2.5}$. Using hourly averaged measured spectra at station PAPA, this gives

a typical random error of 6% if $k_{max}$ corresponds to a 20 m wavelength. The general broadening of the directional spectrum towards high frequency gives a weaker importance of shorter waves (e.g. Peureux et al., 2018).

Given that the wave spectrum varies both along the sea surface and with directions, we have to interpolate either in spectral space (from $\theta$ to $\theta'$) or in physical space ($x$ and $y$). Simulations indicate that variations in physical space are less severe than those in directions, as illustrated on Figure A1, over a Gulf Stream ring where the Stokes drift is enhanced by wind blowing

against the current. We thus estimate $U_{S,R}$ using

$$U_{S,R}(x,y,\theta) \simeq \sum \cos(\theta - \theta') U_{S,1D}(x',y',\theta') \Delta'_\theta, \tag{A2}$$

where the sum is over all directions and the approximation is due to the fact that the location $(x,y)$ is different from $(x',y')$ where the contribution $U_{S,1D}(x',y',\theta')$ is measured. In practice we use the locations $(x',y')$ that are closest to $(x,y)$, separated by a distance $r(\theta')$. The variations of $U_{S,1D}(\theta')$ over the distance $r(\theta')$, which is typically less than 50 km, is mostly due to the effect of currents on waves (Ardhuin et al., 2017b).

In our simulations we have also varied the beam rotation speed, number of beams and number of azimuths per rotation. Because these parameters give different locations of footprints we have used an approximation of the radial Stokes drift $U_{S,R}$ from the map of Stokes drift vectors instead of the full directional spectrum, this is

$$U_{S,R}(x,y,\theta) \simeq \frac{2}{\pi} \sum U_{S,R}(x',y',\theta') \cos(\theta - \theta') \Delta'_\theta, \tag{A3}$$

where the sum is over angles $\theta - \pi/2 < \theta' < \theta + \pi/2$ and the nearest available footprints with these azimuths are taken. Eq. (A3) relies on the assumption of small variability of the vector $(U_S, V_S)$ on the scale of the beam rotation. Indeed, $U_{S,R}(\theta)$ is the projection of $(U_S, V_S)$ in direction $\theta$. If $(U_S, V_S)$ is uniform in space then $U_{S,R} = |U_S, V_S| \cos(\theta - \theta_0)$, with $\theta_0$ the direction of the Stokes drift vector.

We have performed realistic high-resolution simulations in a wide range of conditions: Oregon coast, Gulf Stream, equatorial currents, Fram strait, Agulhas current. Based on all these simulations and assuming a constant $G$, we find that the error $\mathrm{err}_{WB}$ on the estimation of $U_{\mathrm{WB}}$, has a negligible bias and a standard deviation that is of the order of,

$$\mathrm{std}(\mathrm{err}_{\mathrm{WB}}) \simeq \varepsilon G \, \mathrm{std}(U_S) \, r_2/20\mathrm{km} \tag{A4}$$

where $\varepsilon$ is a non-dimensional factor which ranges from 0.10 in the case of the Equator near 23 W, to 0.18 in the Gulf Stream case. $\mathrm{std}(U_S)$ is the standard deviation of the Stokes drift magnitude over the region that contribute to $U_{S,R}$ using eq. (A2).

The distance

$$r_2 = \sqrt{\sum \left[ r(\theta') \cos(\theta - \theta') \right]^2 / N} \tag{A5}$$

is the root mean square distance, over the $N$ cycles with directions $\theta'$ that contribute to the estimation of $U_{S,R}(\theta)$, between the position of the footprint for azimuth $\theta$ and the footprints for $\theta'$ weighted by $\cos(\theta' - \theta)$. Hence, in the open ocean $r_2$ is completely specified by the geometry of the footprints, itself given by the rotation speed of the horn plate and the number of beams. In the case presented here $r_2$ is close to 20 km for the $12°$ beams, and 15 km for the $6°$ beams.

In order to take into account the variability of $G$ and random errors in the estimation of the wave spectrum, we have used an error twice as large as given by eq. (A4).

The error $\mathrm{err}_{WB}$ has smaller scales than both $U_R$ and $U_{S,R}$, as shown in figure A2. The largest errors are associated with current gradients. This suggests that using some knowledge on wave-current interactions could lead to smaller errors.

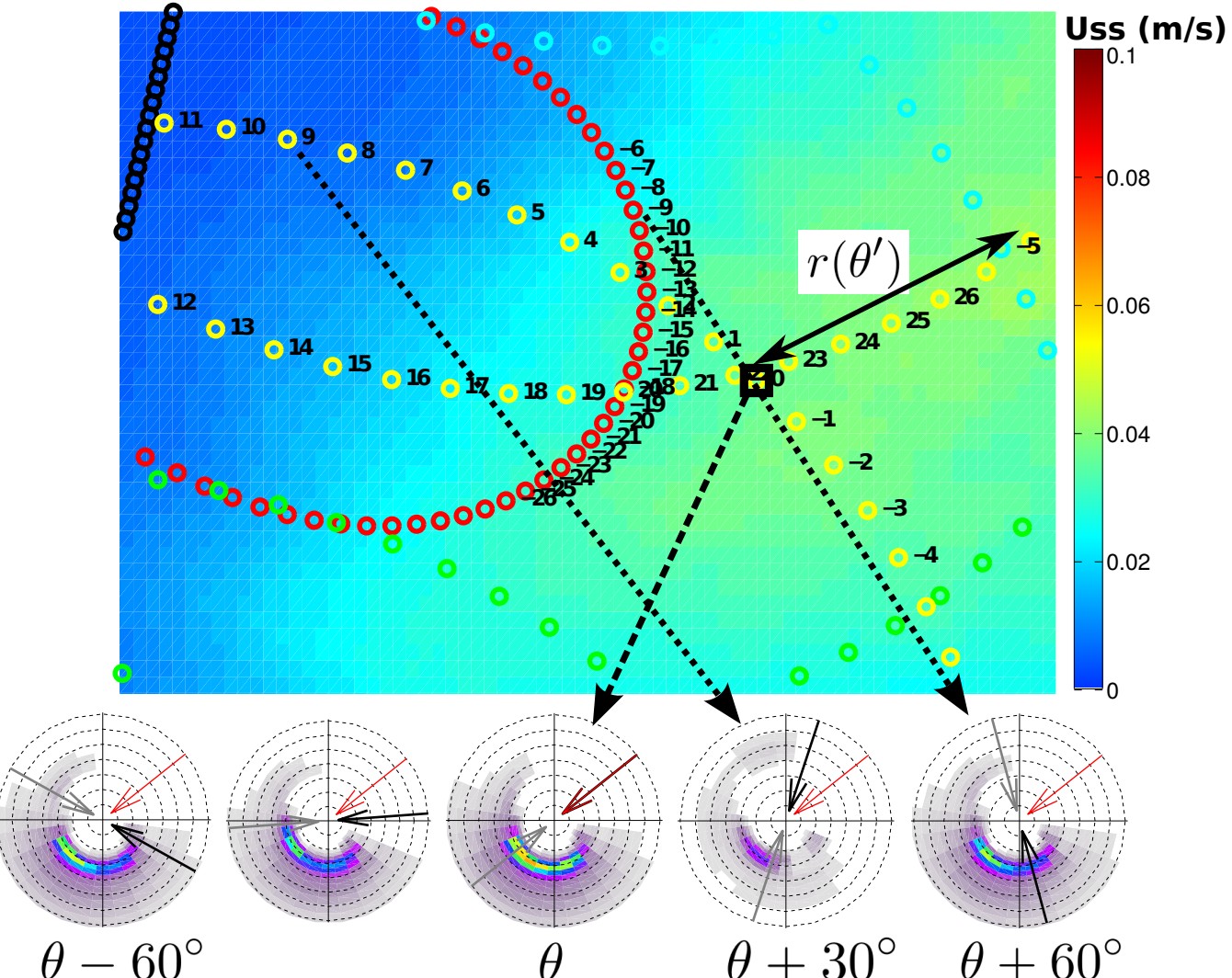

**Figure A1.** Illustration of the use of cycles in azimuth $\theta'$ (circles) for the estimation of the wave bias $U_{WB} = GU_{S,R}$ at the location (black square) of cycle with direction $\theta$. The background color shows the magnitude of $U_S$, and the bottom spectra are at the location of different cycles, with wave energy plotted in the direction from which it arrives. In each spectra the red arrow is direction $\theta$, and the black and grey arrows show $\theta'$ and $\theta' + \pi$. The distance $r(\theta')$ is a source of error.

## Appendix B: Attitude restitution using antenna rotation

The non-geophysical contribution to the Doppler centroid frequency ($f_{NG}$) arises from the acquisition geometry (satellite attitude and instrument pointing) and the platform velocity. This frequency is much higher than the geophysical frequency, and

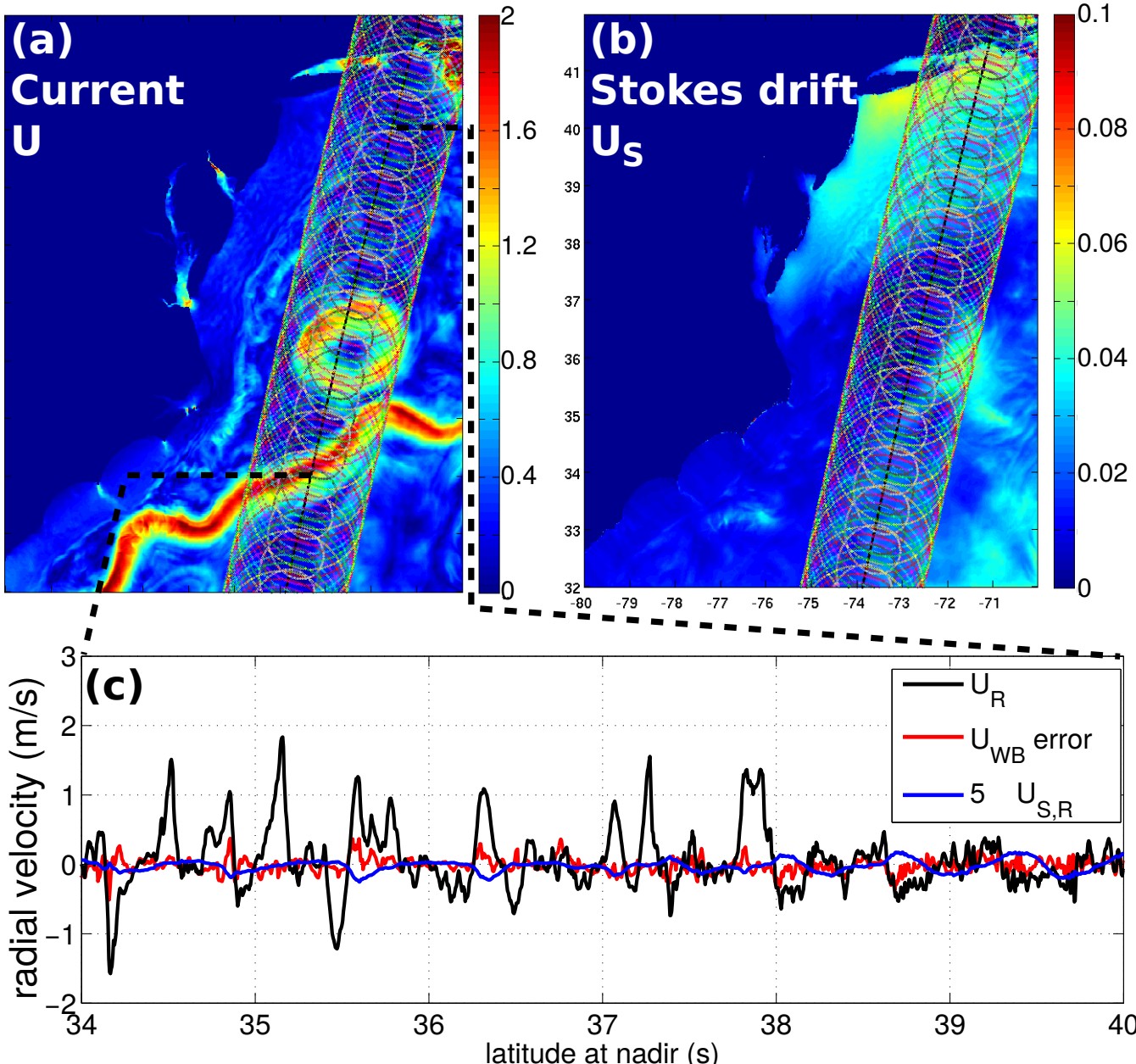

**Figure A2.** Example of (a) current in the Gulf Stream and associated (b) Stokes drift, and (c) errors for the estimation of UWB for beam 8 (12° incidence, green circles in top panels). In this example the standard deviation of $\mathrm{err}_{WB}$ is 10 cm/s, which is of the order of $5U_{S,R}$. Note that the measured geophysical Doppler is $U_{\mathrm{GD}} = U_{\mathrm{WB}} + GU_{S,R}$ with $G \simeq 25$.

it must be estimated carefully. Its theoretical expression is given by Raney (1986)

$$
\begin{aligned}
f_{\mathrm{NG}} \quad = \quad & \frac{2V_{sc}}{\lambda}\sin\gamma\cos\theta \\
& \times \left[1 - \frac{\omega_e}{\omega}(\epsilon\cos\beta\sin\Psi\tan\theta + \cos\Psi)\right] \\
& + \frac{2V_{sc}}{\lambda}e\cos\gamma\frac{\sin(\beta - p)}{\sqrt{1 + e^2 + 2e\cos(\beta - p)}},
\end{aligned}
\tag{B1}
$$

where $\lambda \simeq 8$ mm is the radar wavelength, $V_{sc}$ is the spacecraft velocity, $\gamma$ is the elevation angle, $\theta$ is the azimuth angle, $\epsilon$ is equal to 1 if $\theta \in [0, -\pi]$ and -1 otherwise, $\omega_e$ is the angular rotation rate of the Earth, $\omega$ is the angular rotation rate of the spacecraft, is $\Psi$ the angular position on the orbit, $e$ is the eccentricity of the orbit, $p$ the argument of the perigee.

All the parameters of $f_{\rm NG}$ are well known expect for uncertainties in the azimuth $\theta$ and the elevation $\gamma$ of the radar beam, due the imperfect knowledge of the satellite attitude. As given by eq. (B2), the azimuth and elevation are perturbed by a tilt $\gamma_0$ that is maximum for the azimuth $\theta_0$ relative to the ground track azimuth. These angles are defined in figure B1.

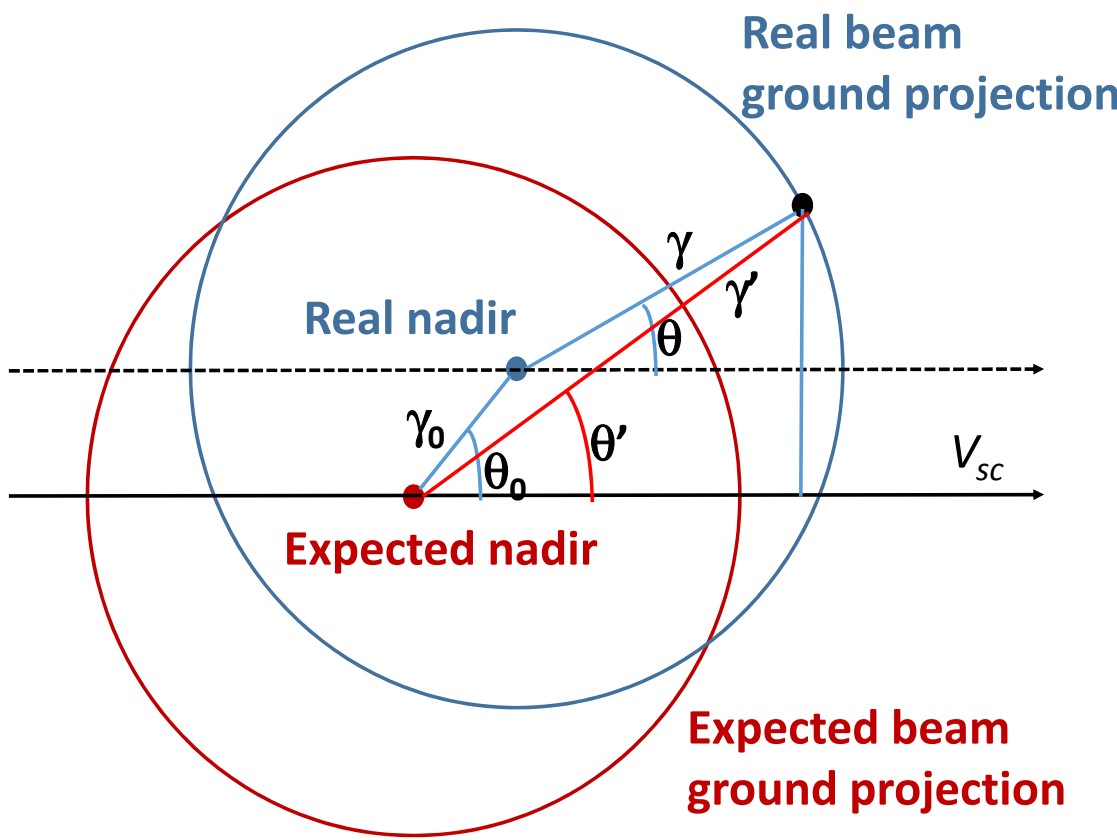

**Figure B1.** Definition of mispointing geometrical parameters $\gamma_0$ and $\theta_0$. We recall that $Z$ is the satellite altitude.

We have assumed that

– the antenna rotation speed is very well known and very stable (it can also be re-estimated after launch on a regular basis),

– the satellite attitude varies slowly (a preliminary requirement is below $10^{-4}$ °/s),

– instrument noise is not correlated in time (pure white noise) and then is uncorrelated with any satellite attitude variation. It is therefore not considered here.

Hence $\gamma_0$ and $\theta_0$ define the difference between the expected and the real nadir position projected on the ground introduced by the satellite attitude misknowledge, so that $\gamma'$ and $\theta'$ are the parameter to be used instead of $\gamma$ and $\theta$ in eq. (B1). It is very important to note that the $\gamma_0$ and $\theta_0$ parameters can vary in time. These time variations describe the satellite attitude misknowledge changes.

Provided that $\gamma_0/\gamma \ll 1$ then

$$
\begin{aligned}
\gamma' &= \gamma + \gamma_0 \cos(\theta - \theta_0) \\
\theta' &= \theta - \gamma_0 \sin(\theta - \theta_0)
\end{aligned}
\tag{B2}
$$

The Doppler shift residue $\delta f_{\mathrm{NG}}(\gamma_0, \theta_0)$ induced by the satellite attitude misknowledge is computed using the eqs. (B1) and (B2).

The situation is better understood by using the fact that $e \ll 1$ and $\omega_e/\omega \ll 1$, leading to simplified equation

$$
\begin{aligned}
\Delta f_{\mathrm{NG}}(\gamma_0, \theta_0) \approx \quad & \frac{2V_{sc}}{\lambda}\gamma_0 \\
& [\cos\gamma\cos\theta\cos(\theta - \theta_0) \\
+ \quad & \sin\gamma\sin\theta\sin(\theta - \theta_0)].
\end{aligned}
\tag{B3}
$$

In our case, $\gamma < 12°$, $\Delta f_{\mathrm{NG}}(\gamma_0, \theta_0)$ is dominated by the term $\cos\gamma\cos\theta\cos(\theta - \theta_0)$ that only contains only twice the beam rotation frequency as shown in figure B2.

For example, looking at the black curve in figure B2, a positive tilt $\gamma_0$ in direction $\theta_0 = 0$ gives the same positive Doppler anomaly when looking forward along the track ($\theta = 0$) when the Doppler is positive and $\gamma' = \gamma + \gamma_0$, and backward $\theta = 180°$ when the Doppler is negative and $\gamma' = \gamma - \gamma_0$.

The pointing knowledge that is provided by the complete system (including SKIM, star strackers, etC.) (around $0.2°$ in elevation and $0.5°$ in azimuth) is not sufficient to get accurate retrieval of $f_{\mathrm{NG}}$. As shown in figure B2, a tilt $\gamma_0$ of only $0.003°$ gives errors in the retrieved radial current speed up to 0.75 m/s. However, compared to Sentinel-1 for which the retrieval of these parameters is complicated by the spacecraft attitude control, in the case of SKIM, we can use the rotation of the antenna beams. The expected variations of the Doppler centroid $f_{\mathrm{DC}} = f_{\mathrm{NG}} + f_{\mathrm{GD}}$ over one or several full rotations can be used to correct for attitude errors. Here we demonstrate the use of a matching algorithm, based on a $f_{\mathrm{NG}}$ model fitting, the amplitude of the velocity residue induced by the satellite attitude misknowledge can be decreased to 3 cm s$^{-1}$.

The capability to estimate $\gamma_0$ and $\theta_0$ from the measured Doppler depends on the contents of the time series of $f_{GD}$ and $f_{NG}$. In cases when currents and Stokes drift are spatially uniform, the dominant geophysical signal $GU_{S,R}$ is dominated by the beam rotation frequency $\omega_b$ (see appendix A), whereas the beam mispointing only contains $2\omega_b$. However, any spatial structure in currents and Stokes drift will produce contributions at $2\omega_b$ that are mixed with the mispointing signal.

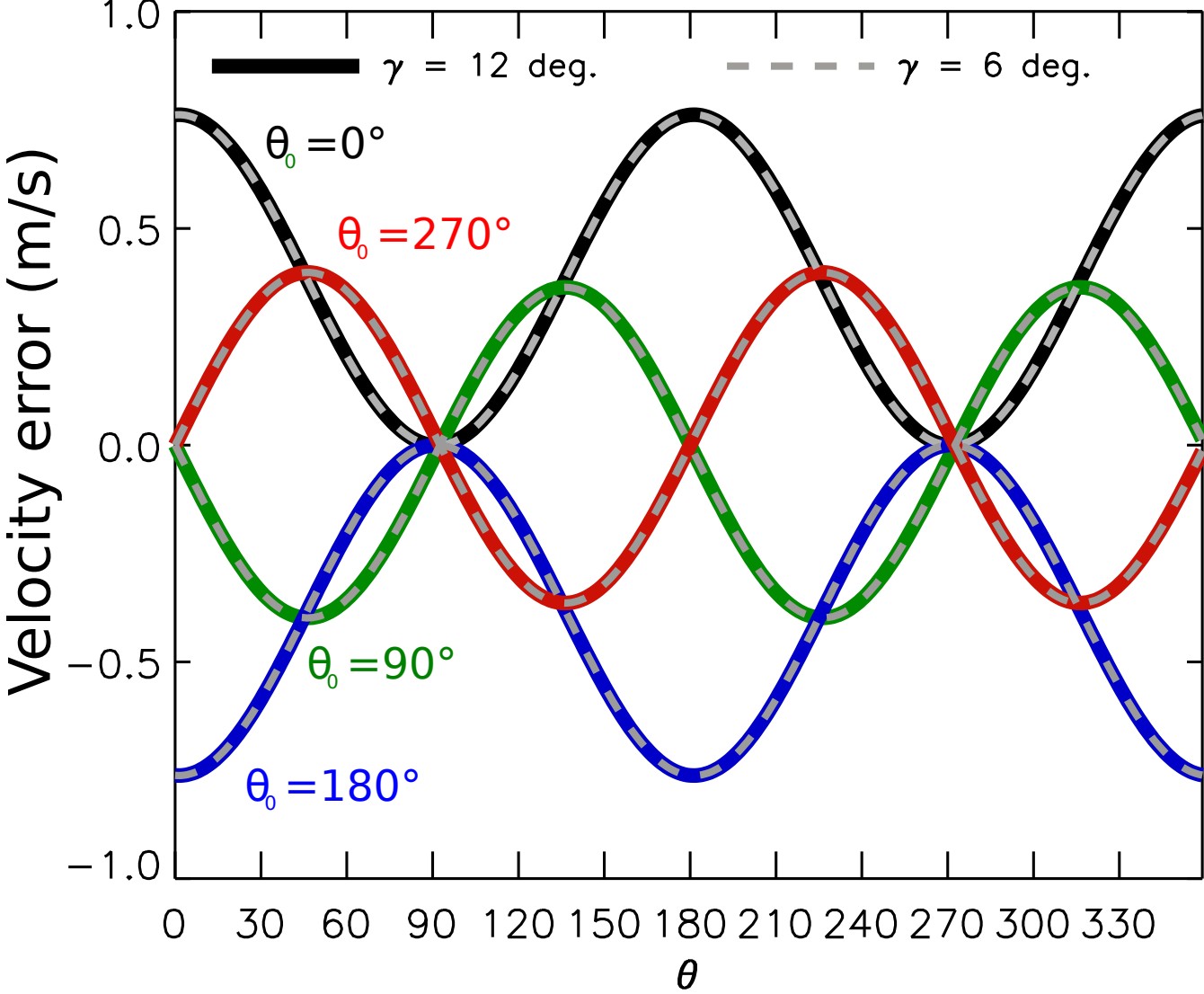

**Figure B2.** Velocity errors induced by a misknowledge $\gamma_0 = 0.003°$ of the satellite attitude, as a function of the azimuth relative to the satellite track, and for four examples of $\theta_0$. The two beams incidence angles are superimposed (thick lines: 12°, thin dashed lines: 6°). These errors are dominated by a contribution at twice the beam rotation frequency $\omega_b$.

Fortunately, there are large differences between $f_{GD}$ and $f_{NG}$ for different incidence angles. At the SKIM angles $\cos \gamma \simeq 1$ and the mispointing errors at 12 and 6° are almost the same (grey dashed lines and thick solid lines are superimposed in Figure B2). In contrast, the geophysical signal is proportional to $\sin \gamma$ and doubles between 6° and 12°.

The next equation that approximates the data model $D_i$ at the sample $i$ is the model used to fit the $\gamma_0$ and $\theta_0$ values in $f_{\mathrm{DC}}$,

$$
\begin{aligned}
D_i = f_{\mathrm{DC},i} - \bar{f}_{\mathrm{NG}} \quad \approx \quad & \sin\gamma_i\left(\cos\theta_i U_p + \sin\theta_i V_p\right) + \\
& A_k \Delta f_{\mathrm{NG},i,x} + \\
& B_k \Delta f_{\mathrm{NG},i,y} + N_i.
\end{aligned}
\tag{B4}
$$

5 In eq. (B4), $f_{\mathrm{DC},i}$ is the observed signal at the sample $i$, $\bar{f}_{\mathrm{NG}}$ is the value of $f_{\mathrm{NG}}$ for the nominal satellite attitude, $(U_{GD,p}, V_{GD,p})$ is the geophysical velocity vector in the pixel $p$ at the sea surface (see below for pixel definition), $(\Delta f_{\mathrm{NG},i,x}, \Delta f_{\mathrm{NG},i,y})$ is the decomposition of the satellite attitude misknowledge at the sample $i$, $N_i$ is the noise at the sample $i$, $A_k = \gamma_k \cos\theta_k$ and $B_k = \gamma_k \sin\theta_k$ are the two parameters to be fitted to characterize the satellite attitude misknowledge expected stable during the period $k$. In the present study $k$ is equal to one full beam rotation.

10 Here the ground speed $(U_p, V_p)$ is taken constant in a given pixel $p$ at the sea surface. The pixel definition uses the `HEALPix` spherical binning (Górski et al., 2005) with $\mathrm{nside} = 256$ equivalent to pixel area of $\approx 14 \times 14 \mathrm{arcmin}^2$, that is 25 by 25 km. This binning provides pixels with the same surface on all the sphere and ease the software writing. Using B4 the difference $R_{i,p}$ is built from all $D_i$ inside a pixel to be independent from any $(U_p, V_p)$ values,

$$
\begin{aligned}
R_{i,p} \quad = \quad & D_i - P.M^{-1}.\begin{pmatrix}\sum_j \sin\gamma_j\cos\theta_j M_j \\ \sum_j \sin\gamma_j\sin\theta_j M_j\end{pmatrix} \\
& - A_k \Delta f_{\mathrm{NG},i,x} \\
& + P.M^{-1}.\begin{pmatrix}\sum_j \sin\gamma_j\cos\theta_j A_k \Delta f_{\mathrm{NG},j,x} \\ \sum_j \sin\gamma_j\sin\theta_j A_k \Delta f_{\mathrm{NG},j,x}\end{pmatrix} \\
& - B_{0,i}\Delta f_{\mathrm{NG},i,y} \\
& + P.M^{-1}.\begin{pmatrix}\sum_j \sin\gamma_j\cos\theta_j B_k \Delta f_{\mathrm{NG},j,y} \\ \sum_j \sin\gamma_j\sin\theta_j B_k \Delta f_{\mathrm{NG},j,y,}\end{pmatrix}
\end{aligned}
\tag{B5}
$$

with

$$
\begin{aligned}
P \quad &= \quad \begin{pmatrix}\sin\gamma\cos\theta_j \\ \sin\gamma\sin\theta_j\end{pmatrix} \\
M \quad &= \quad \begin{pmatrix}\sum_j \sin^2\gamma_j\cos^2\theta_j & \sum_j \sin^2\gamma_j\cos\theta_j\sin\theta_j \\ \sum_j \sin^2\gamma_j\cos\theta_j\sin\theta_j & \sum_j \sin^2\gamma_j\sin^2\theta_j.\end{pmatrix}
\end{aligned}
\tag{B6}
$$

It is important to understand that several rotations are solved together otherwise the matrix $M$ cannot be inverted by lack of redundant information in the same pixel. Thus, set of $\gamma_k$ and $\theta_k$ values are extracted by minimizing the $\chi^2 = \sum_{p,i}\left(R_{i,p}^2\right)$

combining different $\gamma$ beams over several antenna rotation to increase the azimuth sampling. There is still potential correlation between $\Delta f_{\mathrm{NG}}\left(\gamma_0, \theta_0\right)$ and geophysical Doppler frequency shifts because the hypothesis that $(U_p, V_p)$ is always the same is not true in time neither within the signal from different beams. Furthermore, this model is based on a white instrument noise.

Instrument noise does not introduce an average bias as long as it is not correlated in time and not correlated to the linear fit
of $\Delta f_{\mathrm{NG}}\left(\gamma_0, \theta_0\right)$. Further investigations will check the level of the expected correlation between $\Delta f_{\mathrm{NG}}\left(\gamma_0, \theta_0\right)$ and the noise. In our estimates, the reconstructed values of the pair $(\gamma_0, \theta_0)$ are only affected by the correlation between the non-geophysical and geophysical signals.

Simulations have been done to include random drifts of the satellite attitude within $10^{-4}$ °/s variations. Geophysical Doppler contributions $f_{\mathrm{GD}}$ has been computed from surface currents and wave-induced biases estimated from numerical models of the
ocean circulation and waves. The case of Oregon area is illustrated in Fig B3. The standard deviation of the error on Doppler velocity is found to be around 0.04 m/s after cleaning and is related to the correlation between the two simulated values of $\Delta f_{\mathrm{NG}}\left(\gamma_0, \theta_0\right)$ and $f_{\mathrm{GD}}$. The error induced by the satellite attitude misknowledge is time locally higher when the signal has more structures. This advocates to average the determination of these errors on larger data set. This can be possible if the satellite attitude is more stable or the drift follows physical law decoupled from expected $f_{\mathrm{GD}}$ signal.
In conclusion, the case of Oregon area illustrated in Fig B3 shows that the cleaned velocity error is relatively small ($< 0.04$ m/s). The algorithm has been tested on other regions (not illustrated here) with similar but often better results.

*Author contributions.* All authors have contributed to the writing of the paper. The wave bias error analysis was performed by FA, GM and EC analyzed the Doppler centroid estimation error, JMD, CT and EC analyzed the error due to platform attitude. LG and CU combined these errors to produce simulated data that was used by JX for DFS analysis with TOPAZ. The optimal interpolation scheme was adapted by CU
and will be described elsewhere in more details.

*Competing interests.* The authors declare that they have no conflict of interest.

*Acknowledgements.* Support from LabexMer via grant ANR-10-LABX-19-01, and Copernicus Marine Environment Monitoring Service (CMEMS) as part of the Service Evolution program is gratefully acknowledged. Additional support from CNES was given through the VASCO phase 0 study and R&T contracts.

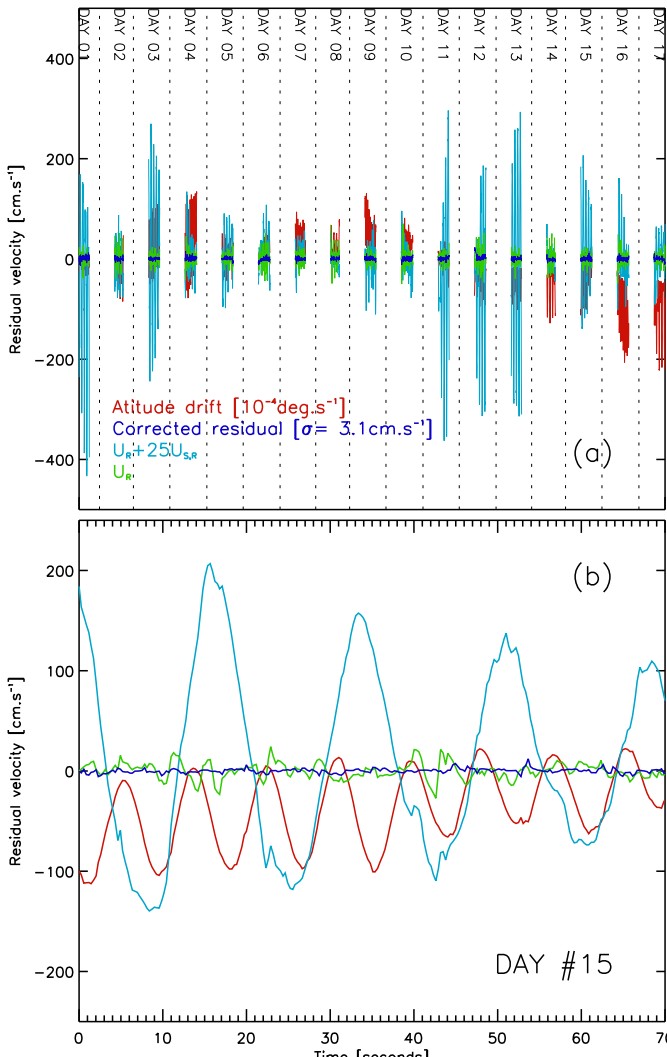

**Figure B3.** Velocity residual error (in cm/s) for the Oregon coast simulations non corrected (red curve) and corrected (blue curve) by removing the fit of the satellite attitude misknowledge combining beams and $8\ (\gamma_0, \theta_0)$ couples by antenna rotation. (a) shows simulated data over one satellite pass for day 21. (b) results for all satellite passes in the domain shown in figure 6. This time frame in September 2014 includes winds speeds up to 22 m/s. For each pass the duration is stretched over 20 hours in order to make it visible. $U_R$ and $U_{R,S}$ signals are also shown for scale. The satellite attitude drifts randomly ($10^{-4}$ °/s). The error induced by the satellite attitude misknowledge is higher when the geophysical signal is stronger and less uniform.

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
