# Peer review of "Measuring currents, ice drift, and waves from space: the Sea Surface KInematics Multiscale monitoring (SKIM) concept"

_Ocean Science, 2017_

## Referee Comment (RC1) · M.A. Bourassa (Referee) · 23 Jan 2018

The paper starts with a wonderful assessment of the spatial temporal sampling capabilities of SKIM, and topics for which it can make substantial contributions. Figure 1 is also extremely effective in describing the spatial-temporal sampling. The advantages and compromises of proposed methods for measuring currents is well described, with the points nicely emphasized in Figure 9. In general, the paper is quite easy to read, with some of the more complex details in appendix A. The clarity of the appendices is not as good as the main body of the text and could be improved. The only large gap is a clear description of the products that are expected.

[Figure]

Major comments:

1) Page 8 & 9: The goal is clearly stated to be the retrieval of the Eulerian velocity. However, the velocity including Stokes drift will be of use for many topics (e.g., oil drift and air-sea fluxes). Will the Stokes drift also be made available?

Minor Comments:

2) Page 2, line 23: ATI provides speeds or more accurately vector components rather than velocities. Similar errors in word usage should be corrected throughout.

3) Page 3, line 7: at (UGD) to the end of the line.

4) Page 5 line, 2, change 'sea surface Us' to 'surface Us'

5) Figure 10 would be better with a sharper color bar

6) The clarity of the caption for fig. A1 should be improved

---

## Referee Comment (RC2) · Anonymous Referee #2 · 23 Jan 2018

This paper describes the concept of a space mission that will utilize the Doppler shift of radar returns to measure the ocean surface velocity, wave parameters, and sea ice drift. However, the small incidence angle of 12 degrees is highly undesirable for meeting the main mission objectives of ocean surface velocity, as well as sea ice drift for the following reasons:

1. Any measurement errors would be amplified by a factor of 1/sin (12 degrees), or ∼4.8.

2. The Doppler shift is heavily contaminated by the radial motions of the waves. The correction for wave bias is very strenuous without much assurance.

[Figure]

3. The narrow swath makes the revisit time at a given location insufficient to sample the high-frequency motions like inertial currents and tidal currents that would overwhelm the low-frequency ocean currents that are the mission's main objectives. As noted in the paper, more than daily revisit will take place only at latitudes higher than 75 degs. The coverage of SKIM over time scales of 1' and 1 " in Fig 1 is way overstretch.

Although the mission would take advantage of the spare parts of the SWIM instrument, they impose the limitation on the incidence angle and therefore are really a wrong choice for meeting the mission science objectives. This is somewhat like using the spare parts of a cheap ordinary car to build a sports car hoping to win the Formula One race. The mission might serve the role of demonstrating the technique, but it is highly unlikely that the mission would advance the knowledge of ocean surface circulation.

––––––––––––––––––––

---

## Referee Comment (RC3) · Anonymous Referee #3 · 25 Jan 2018

**General comments**

Of course this kind of missions are welcome because we badly need a better knowledge of the ocean circulation and, as described, SKIM can provide a good step ahead in this direction. The description is detailed and there is a clear evaluation of the errors implied by the instruments and the related methodology.

**Specific comments**

I am not a specialist of the subject. However, I tried to follow the various arguments and to me, within the just mentioned limitations, they look sound.

[Figure]

One comment concerns the estimate of current below the surface (the proposed instrument measures the surface one). The first author has worked on how the wave propagation velocity depends on the current vertical profile. I wonder if there is any way to use the wave spectrum and surface current information in this respect. However, possibly a counter argument is that the instrument provides only the $(k,\theta)$ spectrum, i.e. a geometrical picture of the situation, with no measured information on its dynamical, implicitly $(f,\theta)$, behaviour.

A possibly more serious one concerns the measurement of the wave spectrum. Figure 5 at page 8 provides a clear perspective of the logical flow of actions and data. It is clear that the availability, hence the measurement, of the $E(k,\theta)$ spectrum is a key point, obviously required, apart from other needs, to estimate the Stokes drift. However, how to measure $E(k,\theta)$ is not detailed and developed enough in my view. The SWIM approach of Danielle Hauser et al (2017) is cited, but SKIM is a different instrument and its performance should be described in more details.

Corrections. No particular corrections (however, I did not read carefully every single line). On the way I spotted three minor typos:

page 18 lines 2 and 16

page 20 line 6

Very good project. Congratulations (and apologies for the short comment),

---

## Author Comment (AC1) · 2 Mar 2018

Please find below a point-by-point reply. In bold are the reviewer comments.

**The clarity of the appendices is not as good as the main body of the text and could be improved. The only large gap is a clear description of the products that are expected.**

We have worked to improve the readability of Appendix A. As for Appendix B it was completely rewritten based on independant and new simulations that include a random (but slow) attitude error compared to the previous random linear trend over 20 s.

[Figure]

**Major comments: 1) Page 8 9: The goal is clearly stated to be the retrieval of the Eulerian velocity. However, the velocity including Stokes drift will be of use for many topics (e.g., oil drift and air-sea fluxes). Will the Stokes drift also be made available?**

The (quasi-) Eulerian velocity will be particularly used for the Level 3 product (multiple swaths with mapping is space and time) because of the rapid variation of the Stokes component which will also be provided in Level 3a products. This is now clarified.

**Minor Comments: 2) Page 2, line 23: ATI provides speeds or more accurately vector components rather than velocities. Similar errors in word usage should be corrected throughout.** We have clarified the use of speed and velocity

**3) Page 3, line 7: at (UGD) to the end of the line.** We are not sure how to understand this comment.

**4) Page 5 line, 2, change 'sea surface Us' to 'surface Us'** This has been corrected

**5) Figure 10 would be better with a sharper color bar** This has been corrected

**6) The clarity of the caption for fig. A1 should be improved** We have corrected the caption as follows: Illustration of the use of cycles in azimuth $\theta$' (circles) for the estimation of the integrated parameters at the location (black square) of cycle with direction $\theta$.

---

## Author Response (AR1)

**Answer to Referees**

The authors would like to thank the reviewers for their comments, which helped improving this manuscript. In particular the introduction was largely rewritten following the important remarks on near-inertial motions, and figure 1 was redrawn. We also completely rewrote appendix B after we found an error in the analysis (we had perturbed the elevation instead of the pointing angles)

Answers to the 3 reviewers are listed below.

**Reviewer #1**
We thank our colleague for the kind and detailed comments and questions.
We also note that we have completely redone the analysis of Appendix B after we realized that we had perturbed the elevation and not the pointing angles, which gave a different periodicity for the error due to attitude misknowledge.

Below a point-by-point reply. In bold are the reviewer comments and our comments follow in normal font.

**The clarity of the appendices is not as good as the main body of the text and could be improved. The only large gap is a clear description of the products that are expected.**

We have worked to improve the readability of Appendix A. As for Appendix B it was completely rewritten based on new simulations, and now gives details of the fitting procedure used to retrieved the beam pointing parameters from the measured Doppler.

**Major comments:**
**1) Page 8 & 9: The goal is clearly stated to be the retrieval of the Eulerian velocity. However, the velocity including Stokes drift will be of use for many topics (e.g., oil drift and air-sea fluxes). Will the Stokes drift also be made available?**

The (quasi-) Eulerian velocity will be particularly used for the Level 3 product (multiple swaths with mapping is space and time) because of the rapid variation of the Stokes component which will also be provided in Level 2 products. This is now clarified.

**Minor Comments:**
**2) Page 2, line 23: ATI provides speeds or more accurately vector components rather than velocities. Similar errors in word usage should be corrected throughout.**

We have clarified the use of "speed" and "velocity"

*{3) Page 3, line 7: at (UGD) to the end of the line.*

We are not sure how to understand this comment.

**{4) Page 5 line, 2, change 'sea surface Us' to 'surface Us'**

This has been corrected

**5) Figure 10 would be better with a sharper color bar**

This has been corrected

**6) The clarity of the caption for fig. A1 should be improved**

We have corrected the caption as follows:
*Illustration of the use of cycles in azimuth θ' (circles) for the estimation of the integrated parameters at the location (black square) of cycle with direction θ.*

**Reviewer #2**

We thank the reviewer for the thoughtful and thought-provoking remarks that have led to important changes in the manuscript.
Below a point-by-point reply. In bold are the reviewer comments and our comments follow in normal font.

**his paper describes the concept of a space mission that will utilize the Doppler shift of radar returns to measure the ocean surface velocity, wave parameters, and sea ice drift. However, the small incidence angle of 12 degrees is highly undesirable for meeting the main mission objectives of ocean surface velocity, as well as sea ice drift for the following reasons:**

**1. Any measurement errors would be amplified by a factor of 1/sin (12 degrees), or about 4.8.**

We understand the reviewer concerns, but the question of errors should always be considered in the context of signal to noise ratio (SNR). Indeed, although the signal of the horizontal current is reduced by a factor 4 when going from, say, 55° to 12°, the noise is reduced by a factor 100 (20 dB) for average wind speeds (6 to 8 m/s, Yurovsky et al. 2016). As a result the SNR is better at 12° compared to larger incidence angles.

**2. The Doppler shift is heavily contaminated by the radial motions of the waves. The correction for wave bias is very strenuous without much assurance.**

Besides noise, measurements indeed contain a wave bias which varies little with incidence angle (from 6 to 20 degrees) because the wave orbital velocities are the same in all directions. It is thus correct that, relative to the current signal, this is amplified by $1/\sin(\theta i)$.

In the end, the wave bias is of the same order of magnitude for Ka band at 12° as it is for C band at 23°, which corresponds to the wave mode data on Envisat used by Chapron et al. (2005) and Collard et al. (2008), and from which it was possible to measure the equatorial currents very clearly using a proxy based on modeled wind. Here we wish to correct the wave bias more precisely so that we do not need to average the data over many passes. A preliminary algorithm is described here. We expect that it will be perfected in the coming years.

**3. The narrow swath makes the revisit time at a given location insufficient to sample the high-frequency motions like inertial currents and tidal currents that would overwhelm the low-frequency ocean currents that are the mission's main objectives. As noted in the paper, more than daily revisit will take place only at latitudes higher than 75 degs.**

The mission's main objective is to measure currents, whatever their variability and nature. Indeed, surface currents contain many contributions (tides, near-inertial motions …), that -- in the case of coastal regions -- are very well revealed by HF radars (e.g. Kim and Kosro 2013, Kim 2014) and driter data (e.g. Poulain et al. 2013, Elipot and Lumpkin 2016).  The relative magnitude of these inertial currents is highly variable and can range from 10% to 60% (Poulain et al. 2013 have a low ratio in the Mediterranean, Kim and Kosro have a higher ratio off the U.S. West coast, other places are tide dominated, such as the French continental shelf, see Ardhuin et al. 2009).

The sampling is certainly not ideal to resolve all these. This question raises the issue of how the data will be used. It is the same question with the diurnal aliasing in QuikScat wind measurements, and the relative variability of currents on a 3-day time scale and at 30 km resolution may be comparable to the variability of winds on a 12h time scale and at 25 km resolution. We have thus added the following sentences in the introduction:

*As detailed below, the Sea Surface KInematics Multiscale monitoring (SKIM) mission, propose to use map surface waves and currents with 6-km footprints with a 4 m resolution in range. These footprints are distributed across a 270 km wide swath, but do not cover the entire swath, leaving a gap between the features smaller than 6 km resolved with a footprint, and the features larger than 20 km fully mapped across the swath.*

*As the ocean is viewed in less than 1 minute during a single pass, the observed scene is basically a snapshot in which many ocean processes are aliased. Only those current features that vary on time scales of several days, or that have a constant phase and amplitude such as tides, can be measured without ambiguity. Evidence from High Frequency radars in coastal areas suggests that even near-inertial motions are coherent over time scales as large as 6 days at mid-latitudes \cite{Kim&Kosro2013}. Hence measured currents, even if every 3 days only, can provide useful constraints on the ocean circulation.*

Another issue raised by the question is which contributions are least predictable and thus most useful in a data assimilative prediction system. We note that good results have already been obtained for tides and inertial motions in the past (Stammer et al. 2014, Jing et al. 2014).

Stammer, D. et al. (2014). Reviews of Geophysics Accuracy assessment of global barotropic ocean tide models. Reviews of Geophysics, 52, 243–282. http://doi.org/10.1002/2014RG000450.

Jing, Z., Wu, L., & Ma, X. (2015). Improve the simulations of near-inertial internal waves in the ocean general circulation models. Journal of Atmospheric and Oceanic Technology, 32(10), 1960–1970. http://doi.org/10.1175/JTECH-D-15-0046.1

**The coverage of SKIM over time scales of 1' and 1'' in Fig 1 is way overstretch.**

We agree and figure 1 has been completely redrawn and completed, and clarified. The meaning of the scales in Figure 1 is now clarified in the caption: 1' is the maximum time lag between two views of the same ocean region over one single pass: flying at 7 km/s with a 270 km swath diameter gives 38 s maximum time lag. And the acquisition frequency if around 4 Hz (dt = 0.23 s) for all beams. Hence these two time scales define the range of time within a "snapshot".

**Although the mission would take advantage of the spare parts of the SWIM instrument, they impose the limitation on the incidence angle and therefore are really a wrong choice for meeting the mission science objectives. This is somewhat like using the spare parts of a cheap ordinary car to build a sports car hoping to win the Formula One race. The mission might serve the role of demonstrating the technique, but it is highly unlikely that the mission would advance the knowledge of ocean surface circulation.**

"Spare parts" is an exaggeration: SKIM is not using "spare parts" but builds on an existing design, with Ka band instead of Ku, a larger reflector (1.2 m instead of 0.8 m), and, most importantly Doppler capability which requires a very high PRF because of geometrical decorrelation. As for a "race", SKIM is now in phase A with ESA and this will certainly help raising the profile of all proposals for measuring ocean circulation with Doppler. The fact that some components inheriting from previous missions certainly helped fitting in the tight budget of ESA Earth Explorer 9, not very different from the budget of SMOS. Clearly technological readiness level (TRL) was a big issue for this Earth Explorer 9. The incidence angle of 12° gives a 270 km swath for a 690 km orbit. This yields a revisit time of 3 days at mid latitudes that is consistent with the dynamical evolution time of 60 km wavelength patterns in the mesoscale field.

Given the preliminary result of Envisat we are thus confident that SKIM can deliver very useful data on ocean dynamics, in particular for tropical currents, mapping the equatorial divergence, tropical instability waves, but also the global mesoscale and inertial oscillations.

**Reviewer 3**

**One comment concerns the estimate of current below the surface (the proposed instrument measures the surface one). The first author has worked on how the wave propagation velocity depends on the current vertical profile. I wonder if there is any way to use the wave spectrum and surface current information in this respect. However, possibly a counter argument is that the instrument provides only the (k,θ) spectrum, i.e. a geometrical picture of the situation, with no measured information on its dynamical, implicitly $(f,\theta)$, behaviour.}**

A larger dwell time than the 30~ms focused on a single footprint would be needed for going into a frequency-wavenumber analysis. Still, filtering data at a given wavenumber with the Delta-K technique (e.g. Alpers & Hasselmann)  may - theoretically - allow the analysis of this shear by measuring the Doppler shift for different Delta-Ks, similar to what was done by Shrira et al. (2001) for HF radars. We have now clarified on page 8 (lines 19-24) the effect of a vertical shear on the measured velocities and our opinion that measuring the shear does not appear feasible.

**A possibly more serious one concerns the measurement of the wave spectrum. Figure 5 at page 8 provides a clear perspective of the logical flow of actions and data. It is clear that the availability, hence the measurement, of the E(k,θ) spectrum is a key point, obviously required, apart from other needs, to estimate the Stokes drift. However, how to measure E(k,θ) is not detailed and developed enough in my view.**

**The SWIM approach of Danielle Hauser et al (2017) is cited, but SKIM is a different instrument and its performance should be described in more details.**

We have added a few details, and we also refer the reader to Nouguier et al. (2018). Without doubling the length of the paper it would be hard to cover the topic. We thus conclude the introduction section with
*The present paper focuses on currents, and a detailed description of wave measuring capabilities with SKIM will be given elsewhere.*

**page 18 lines 2 and 16**
These sentences have been re-written.

**page 20 line 6**
This sentence has been rewritten.

[revised manuscript text omitted]

$$B_k \Delta f_{\text{NG},i,y} + N_i. \tag{B4}$$

In eq. (B4), $f_{\text{DC},i}$ is the observed signal at the sample $i$, $\bar{f}_{\text{NG}}$ is the value of $f_{\text{NG}}$ for the nominal satellite attitude, $(U_{GD,p}, V_{GD,p})$ is the geophysical velocity vector in the pixel $p$ at the sea surface (see below for pixel definition), $(\Delta f_{\text{NG},i,x}, \Delta f_{\text{NG},i,y})$ is the decomposition of the satellite attitude misknowledge at the sample $i$, $N_i$ is the noise at the sample $i$, $A_k = \gamma_k \cos\theta_k$ and

25 $B_k = \gamma_k \sin\theta_k$ are the two parameters to be fitted to characterize the satellite attitude misknowledge expected stable during the period $k$. In the present study $k$ is equal to one full beam rotation.

Here the ground speed $(U_p, V_p)$ is taken constant in a given pixel $p$ at the sea surface. The pixel definition uses the `HEALPix` spherical binning (Górski et al., 2005) with $\text{nside} = 256$ equivalent to pixel area of $\approx 14 \times 14 \text{arcmin}^2$, that is 25 by 25 km. This binning provides pixels with the same surface on all the sphere and ease the software writing. Using B4 the difference

5 $R_{i,p}$ is built from all $D_i$ inside a pixel to be independent from any $(U_p, V_p)$ values,

$$R_{i,p} \quad = \quad D_i - P.M^{-1}. \quad \begin{pmatrix} \sum_j \sin\gamma_j \cos\theta_j M_j \\ \sum_j \sin\gamma_j \sin\theta_j M_j \end{pmatrix}$$

$$- \quad A_k \Delta f_{\text{NG},i,x}$$

$$+ \quad P.M^{-1}. \quad \begin{pmatrix} \sum_j \sin\gamma_j \cos\theta_j A_k \Delta f_{\text{NG},j,x} \\ \sum_j \sin\gamma_j \sin\theta_j A_k \Delta f_{\text{NG},j,x} \end{pmatrix}$$

$$- \quad B_{0,i} \Delta f_{\text{NG},i,y}$$

[revised manuscript text omitted]

Stokes, G. G.: On the theory of oscillatory waves, Trans. Camb. Phil. Soc., 8, 441–455, 1849.

20  Sudre, J., Maes, C., and Garçon, V.: On the global estimates of geostrophic and Ekman surface currents, Limnology and Oceanography: Fluids and Environments, 3, 1–20, doi:10.1215/21573689-2071927, 2013.

Sutherland, P. and Gascard, J. C.: Airborne remote sensing of ocean wave directional wavenumber spectra in the marginal ice zone, Geophys. Res. Lett., 43, 4659–4664, doi:10.1002/grl.53444, 2016.

The SKIM Team: Sea surface KInematics Multiscale monitoring, full proposal for ESA EE9, Tech. rep., Laboratoire d'Océanographie

25  Physique et Spatiale, Brest, France, doi:10.13140/RG.2.2.18902.86084/1, prepared for European Space Agency, 192 pp., 2017.

Thomson, J., D'Asaro, E. A., Cronin, M. F., Rogers, W. E., Harcourt, R. R., and Shcherbina, A.: Waves and the equilibrium range at Ocean Weather Station P, J. Geophys. Res., 118, 595–5962, doi:10.1002/2013JC008837, 2013.

Ubelmann, C., Cornuelle, B., and Fu, L.-L.: Dynamic Mapping of Along-Track Ocean Altimetry: Method and Performance from Observing

605  System Simulation Experiments, J. Atmos. Ocean Technol., 33, 1691–1699, doi:10.1175/JTECH-D-15-0163.1, 2016.

van Sebille, E., Wilcox, C., Lebreton, L., Maximenko, N., Hardesty, B. D., van Franeker, J. A., Eriksen, M., Siegel, D., Galgani, F., and Law, K. L.: A global inventory of small floating plastic debris, Environ. Res. Lett., 10, 124 006, doi:10.1088/1748-9326/10/12/124006, 2015.

Wollstadt, S., López-Dekker, P., De Zan, F., and Younis, M.: Design Principles and Considerations for Spaceborne ATI SAR-Based Observations of Ocean Surface Velocity Vectors, IEEE Trans. on Geosci. and Remote Sensing, 99, 1–20, doi:10.1109/TGRS.2017.2692880,

610  2016.

Xie, J., Bertino, L., Counillon, F., Lisæter, K. A., and Sakov, P.: Quality assessment of the TOPAZ4 reanalysis in the Arctic over the period 1991–2013, Ocean Science, 13, 123–144, doi:10.5194/os-13-123-2017, http://www.ocean-sci.net/13/123/2017/, 2017.

Yurovsky, Y. Y., Kudryavtsev, V. N., Grodsky, S. A., and Chapron, B.: Normalized Radar Backscattering Cross-section and Doppler Shifts of the Sea Surface in Ka-band, in: Proceedings of the Progress In Electromagnetics Research Symposium (PIERS), May 2017, St Petersburg,

615  Russia, IEEE, 2017.

Yurovsky, Y. Y., Kudryavtsev, V. N., Chapron, B., and Grodsky, S. A.: Sea surface kinematics from near-nadir radar measurements, IEEE Trans. on Geosci. and Remote Sensing, p. in press, doi:10.1109/TGRS.2017.2787459, 2018.

Zrnic, D. S.: Spectral Moment Estimates from Correlated Pulse Pairs, IEEE Trans. Aero. Electronic Sys., 13, 344–354, doi:10.1109/TAES.1977.308467, 1977.